# Cooperative regulation of PBI1 and MAPKs controls WRKY45 transcription factor in rice immunity

Kota Ichimaru[1,7], Koji Yamaguchi[1,7], Kenichi Harada[2,7], Yusaku Nishio[1], Momoka Hori[1], Kazuya Ishikawa[1,3], Haruhiko Inoue[4], Shusuke Shigeta[1], Kento Inoue[1], Keita Shimada[1], Satomi Yoshimura[1], Takumi Takeda[3], Eiki Yamashita [2], Toshimichi Fujiwara [2], Atsushi Nakagawa [2], Chojiro Kojima [2,5✉] & Tsutomu Kawasaki [1,6✉]

The U-box type ubiquitin ligase PUB44 positively regulates pattern-triggered immunity in rice. Here, we identify PBI1, a protein that interacts with PUB44. Crystal structure analysis indicates that PBI1 forms a four-helix bundle structure. PBI1 also interacts with WRKY45, a master transcriptional activator of rice immunity, and negatively regulates its activity. PBI1 is degraded upon perception of chitin, and this is suppressed by silencing of *PUB44* or expression of *XopP*, indicating that PBI1 degradation depends on PUB44. These data suggest that PBI1 suppresses WRKY45 activity when cells are in an unelicited state, and during chitin signaling, PUB44-mediated degradation of PBI1 leads to activation of WRKY45. In addition, chitin-induced MAP kinase activation is required for WRKY45 activation and PBI1 degradation. These results demonstrate that chitin-induced activation of WRKY45 is regulated by the cooperation between MAP kinase-mediated phosphorylation and PUB44-mediated PBI1 degradation.

[1] Department of Advanced Bioscience, Graduate School of Agriculture, Kindai University, Nakamachi, Nara 631-8505, Japan. [2] Institute for Protein Research, Osaka University, 3-2 Yamadaoka, Suita, Osaka 565-0871, Japan. [3] Iwate Biotechnology Research Center, Iwate 024-0003, Japan. [4] Plant Function Research Unit, Division of Plant and Microbial Sciences, Institute of Agrobiological Sciences, National Agriculture and Food Research Organization (NARO), Kannondai 2-1-2Ibaraki Tsukuba 305-8602, Japan. [5] Graduate School of Engineering, Yokohama National University, 79-5 Tokiwadai, Hodogaya-ku, Yokohama 240-8501, Japan. [6] Agricultural Technology and Innovation Research Institute, Kindai University, Nakamachi, Nara 631-8505, Japan. [7]These authors contributed equally: Kota Ichimaru, Koji Yamaguchi, Kenichi Harada. ✉email: kojima-chojiro-xk@ynu.ac.jp; t-kawasaki@nara.kindai.ac.jp

Plants have developed sophisticated immune systems to defend against pathogens and to restrict pathogen proliferation. Immune responses are initiated when plasma membrane-localized pattern-recognition receptors (PRRs) are activated by microbe-associated molecular patterns[1]. PRRs are receptor-like kinases with cytoplasmic kinase domains or receptor-like proteins with short cytoplasmic tails. Receptor-like kinases and receptor-like proteins possess extracellular ectodomains that directly bind to their ligands. Ligand perception by PRRs rapidly activates intracellular mitogen-activated protein (MAP) kinases and the production of reactive oxygen species[2–5]. These then activate a series of immune responses, including the production of antimicrobial compounds and reinforcement of the plant cell wall[6]. This kind of immunity is referred to as pattern-triggered immunity. To inhibit these host immune responses, pathogens deliver a variety of effectors into host cells[7]. The effectors target important host immune factors, including PRRs and downstream signaling factors, and suppress their activity, allowing the pathogens to colonize the plant. To overcome effector-mediated inhibition of host immunity, plants have developed an effector-induced immune system referred to as effector-triggered immunity. This is mediated by intracellular immune receptors that are members of the nucleotide-binding leucine-rich repeat (NB-LRR) protein family. These proteins are structurally similar to the NB oligomerization domain-like receptors (NLRs) of mammals[1], and therefore, they are commonly referred to as plant NLRs[8].

Chitin and peptidoglycan are typical microbe-associated molecular patterns derived from fungal and bacterial cell walls, respectively. In rice, chitin and peptidoglycan are recognized by the lysin motif (LysM) receptor-like proteins CEBiP and LYP4/6, respectively[9–11]. Upon ligand perception, CEBiP and LYP4/6 interact with the LysM receptor-like kinase OsCERK1 at the plasma membrane, and then OsCERK1 transmits the signal to intracellular components[11,12]. Thus, OsCERK1 is a common factor regulating both fungal chitin- and bacterial peptidoglycan-triggered immunity. In response to the chitin signal, the cytoplasmic kinase domain of OsCERK1 phosphorylates the rice receptor-like cytoplasmic kinase OsRLCK185, which then phosphorylates MAP kinase kinases such as MAPKKK11, MAPKKK18, and MAPKKK24/MAPKKKε. These kinases then trigger the intracellular activation of the MAP kinases MPK3 and MPK6[13–15]. The activated MAP kinases (MAPKs) induce robust immune responses by phosphorylating downstream immune factors including transcription factors.

Robust immune responses are mediated via the transcriptional regulation of immune-related genes by different types of transcription factors. The WRKY transcription factors are one of the transcription factor families that play important roles in plant immunity[16]. WRKY45 is a major transcriptional activator of the immune response in rice. Enhanced expression of WRKY45 confers resistance to bacterial blight caused by Xanthomonas oryzae pv. oryzae (Xoo) and fungal blast diseases caused by Magnaporthe oryzae[17]. The level of WRKY45 protein is regulated by the ubiquitin-proteasome system[18], and its activity is regulated via phosphorylation by the MAPK MPK6[19]. In fact, the phosphomimic mutant of WRKY45 possesses enhanced transcriptional activity. In tobacco and Arabidopsis, the DNA-binding activities of WRKYs are regulated via MAPK-mediated phosphorylation[20,21].

WRKY45 has also been identified as a key regulator that participates in durable, broad-spectrum blast resistance mediated by the Panicle blast 1 (Pb1) gene. Pb1 encodes an NB-LRR protein with an N-terminal coiled coil (CC) domain (CC-NLR)[22]. Pb1 interacts with and stabilizes WRKY45 through its CC domain, probably by inhibiting proteasome-mediated degradation of WRKY45, and this enhances the immune responses mediated by WRKY45.

WRKY45 appears to be autoregulated, because artificial expression of WRKY45 induces expression of the endogenous WRKY45 gene[23]. Since elevated levels of WRKY45 mRNA cause reductions in growth[24,25], the transcriptional activity of WRKY45 must be suppressed in the absence of pathogens. This suggests the existence of a mechanism that inhibits WRKY45 transcriptional activity under unelicited condition.

Protein ubiquitination is an important post-translational modification process that marks target proteins for degradation via the proteasome[26]. Increasing evidence indicates that ubiquitination plays important roles in a variety of plant cellular processes including immunity, hormone responses, and development[27]. The ubiquitination reaction directs the covalent conjugation of conserved ubiquitin molecules onto protein substrates through the sequential activities of a ubiquitin-activating enzyme (E1), a ubiquitin-conjugating enzyme (E2), and a ubiquitin ligase (E3). Plant E3 ligases are classified into three classes: HECT (homologous to E6-associated protein C-terminus), RING finger, and U-box[26].

Recent studies have revealed that plant U-box type ubiquitin ligases (plant U-box proteins; PUBs) are involved in the positive and negative regulation of defense responses against a variety of pathogens[28]. The tobacco PUB protein CMPG1 and its wheat homolog contribute positively to immune responses[29,30]. In contrast, the three closely related Arabidopsis PUBs AtPUB22, AtPUB23, and AtPUB24 negatively regulate pattern-triggered immunity[31]. AtPUB22 ubiquitinates Exo70B2, a subunit of the exocyst complex that mediates vesicle tethering during exocytosis[32]. AtPUB12 and AtPUB13 ubiquitinate the flagellin receptor FLS2, leading to its degradation and the attenuation of signaling by flagellin, which is another microbe-associated molecular pattern[33,34]. AtPUB25 and AtPUB26 also negatively regulate flagellin signaling by marking the FLS2-associated receptor-like cytoplasmic kinase BIK1 for degradation[35]. AtPUB12 and AtPUB13 also interact with the LysM receptor-like kinases CERK1 and LYK5, respectively, to negatively regulate pattern-triggered immunity[36,37]. The rice PUB protein SPL11 is the closest homologue of AtPUB12 and AtPUB13, and it negatively regulates cell death in rice[38]. Thus, PUBs regulate multiple steps in plant immunity.

Xoo uses a type III secretion system to deliver effectors to the plant cell[39]. XopP, one of the Xoo type III effectors, targets the rice U-box type ubiquitin ligase PUB44[40] and inhibits its ubiquitin ligase activity by interacting with its U-box domain. Silencing of PUB44 suppresses peptidoglycan- and chitin-induced immunity and resistance to Xoo[40], indicating that PUB44 functions downstream of OsCERK1. However, PUB44 is not involved in the activation of MAPKs that is regulated by OsRLCK185[40]. This suggests that PUB44 plays a positive role in an immune pathway that is independent of, or downstream of OsRLCK185.

To understand the molecular mechanism by which PUB44 regulates pattern-triggered immunity, we screened for proteins that interacted with PUB44. We identified PBI1 (PUB44-INTERACTING 1), which belongs to a protein family with the DUF1110 domain. PBI1 exhibits a unique tertiary structure composed of a four-helix bundle. This structure is similar to those of the CC domains of CC-NLR immune receptors. Chitin treatment induces PUB44 phosphorylation and the PUB44-dependent degradation of PBI1, suggesting that PUB44 may induce immune responses through the degradation of PBI1.

PBI1 interacts with WRKY45 and negatively regulates its activity. Therefore, it appears that PBI1 keeps WRKY45 in an inactive state in the absence of pathogen attack, and that PUB44-mediated degradation of PBI1 activates immunity via the de-

suppression of WRKY45. Knockout mutations of *PBI1* increase the protein levels of WRKY45, possibly by releasing it from the PBI1-mediated inhibition of WRKY45 autoregulation. In addition, the degradation of PBI1 is greatly reduced in the *mapkkk11/mapkkk18* double mutant, suggesting that the MAPK pathway also regulates the chitin-induced degradation of PBI1. Thus, it is likely that PUB44 and the MAPKs cooperatively regulate WRKY45 via PBI1 in rice immunity.

## Results

**PBI1 interacts with PUB44**. PUB44 has ubiquitin ligase activity and positively regulates pattern-triggered immunity[40]. This suggests that PUB44 controls immune responses through ubiquitination of immune factors that interact with its ARM domain. To identify substrates for PUB44, we screened proteins that interact with the ARM domain using a yeast two-hybrid system. We identified the product of gene *Os01g0156300* (Rice Annotation Project Database) as a candidate, and named the gene *PUB44-INTERACTING PROTEIN 1* (*PBI1*). We also isolated two other candidates by initial two-hybrid screening (Supplementary Fig. 1). However, we could not confirm the interaction of them with PUB44 using any other methods than the yeast two-hybrid assay. BLAST database searches revealed that PBI1 contains a DUF1110 domain with unknown function. Rice contains three additional DUF1110-containing proteins (PBI2–PBI4; Supplementary Fig. 2), and a phylogenic analysis indicated that PBI1 exhibits highest similarity to PBI2 (72% identity at the amino acid level) (Fig. 1a). *PBI2* is located adjacent to *PBI1* on chromosome 1, with an interval of approximately 1 kb between the coding regions.

We carried out yeast two-hybrid assays to look for interactions between PUB44 and each member of the PBI family. PUB44 interacts with PBI1 and PBI2, but not with PBI3 or PBI4 (Fig. 1b). Rice PUB45 and PUB46 are the closest homologs of PUB44, but they do not have the PUB44-specific U-box sequence that is targeted by the XopP effector[40]. Our two-hybrid analyses indicated that neither PBI1 nor PBI2 interacts with PUB45 or PUB46 (Fig. 1c).

To identify which domains of PUB44 bind to PBI1 and PBI2, we used three constructs containing the U-box and/or ARM domains, separated by a 101 aa linker domain (Fig. 1d). These constructs were designated as PUB44$^{1-452}$, PUB44$^{1-203}$, and PUB44$^{102-452}$. A two-hybrid experiment indicated that PBI1 interacts with PUB44$^{102-452}$ but not PUB44$^{1-203}$ (Fig. 1d). This result is consistent with the fact that PBI1 was isolated using the ARM domain. In contrast, PBI2 interacts with PUB44$^{1-203}$ but not PUB44$^{102-452}$, indicating that PBI2 interacts with the U-box domain. Thus, PBI1 and PBI2 interact with different domains of PUB44, even though they are highly homologous with one another (Supplementary Fig. 2). The fact that PBI1 (but not PBI2) interacts with the ARM domain suggests that PBI1 may be the substrate of PUB44. Therefore, we focused on PBI1 in further studies.

To examine in vivo interaction between PUB44 and PBI1, we carried out co-immunoprecipitation assay. T7-tagged PUB44 and green fluorescent protein (GFP)-tagged PBI1 were transiently expressed in rice protoplasts. PUB44-T7 was co-immunoprecipitated with GFP-PBI1 (Fig. 1e). In addition, we used a split NanoLuc-luciferase complementation system[41]. PUB44 was fused to the Large BiT (LgBiT) of NanoLuc luciferase, and PBI1 was fused to the Small BiT (SmBiT). These constructs were transfected into rice protoplasts, and then total luciferase activities were measured. Co-expression of PUB44 and PBI1 displayed stronger luciferase activity as compared with a control (Fig. 1f). These results indicate that PUB44 and PBI1 interact

with each other in vivo. Furthermore, an in vitro binding assay was used to examine direct interaction between PUB44 and PBI1. Recombinant proteins of glutathione S-transferase (GST)-fused PUB44 and PBI1 were prepared using an *E. coli* protein expression system[42]. The GST-pull down assay indicated that PUB44 directly interacts with PBI1 (Fig. 1g). Since PUB44 is involved in chitin- and peptidoglycan (PGN)-induced immunity[40], we asked whether these PAMPs affect the interaction between PUB44 and PBI1. The protein level of PUB44-T7 co-immunoprecipitated with GFP-PBI1 was increased by treatment with (GlcNAc)$_7$, a chitin oligomer with a degree of polymerization of 7 (Fig. 1h) or PGN (Fig. 1i), suggesting that perception of chitin and PGN enhances the interaction between PUB44 and PBI1.

**Decreases in PBI1 levels depend on PUB44 in response to chitin**. The interaction between PBI1 and the ARM domain of PUB44 suggests that the protein level of PBI1 may be regulated via a ubiquitin-proteasome pathway. To determine the protein level of PBI1 during the chitin and PGN responses, we used recombinant PBI1 protein to raise an antibody (α-PBI1) that specifically recognizes PBI1 (Supplementary Fig. 3a, b). We treated rice suspension-cultured cells with chitin and carried out immunoblot analysis with α-PBI1. The PBI1 protein levels decreased gradually after chitin treatment (Fig. 2a). We then used quantitative real-time PCR to analyze *PBI1* transcript levels, and found that *PBI1* expression was not significantly altered after chitin treatment (Supplementary Fig. 4a). Therefore, it is likely that the chitin-induced reduction in PBI1 occurs at the protein level. We also tested whether treatment of peptidoglycan (PGN) induces the PBI1 degradation. PGN elicitation strongly induced the PBI1 degradation (Fig. 2b). In addition, treatment with the proteasome inhibitor MG132 induced the accumulation of PBI1 (Fig. 2c), indicating that the PBI1 protein concentration is regulated via proteasome-mediated protein degradation.

To test the possibility that PUB44 may participate in the chitin-induced degradation of PBI1, we used two *PUB44* knock down (*PUB44-kd*) cell lines in which the levels of *PUB44* mRNA were greatly reduced by the RNAi method[40]. An immunoblot with α-PBI1 indicated that the PBI1 protein level was reduced in the *PUB44-kd* cell lines (Fig. 2d and Supplementary Fig. 4b), whereas the *PBI1* mRNA levels was slightly higher in the *PUB44-kd* cell than that in wild type (Supplementary Fig. 4c). It seems that PUB44 might be required to maintain steady state level of PBI1. Chitin treatment failed to further reduce the levels of PBI1 protein in both *PUB44-kd* cell lines (Fig. 2d and Supplementary Fig. 4b), whereas the *PBI1* expression in the *PUB44-kd* cell was not changed by chitin treatment (Supplementary Fig. 4d). These results indicate that silencing of *PUB44* inhibited chitin-induced PBI1 degradation. To further confirm these results, we produced the *PUB44 RNAi-ver2* construct using different region (+1,204 bp to +1,562 bp) of *PUB44* cDNA clone (accession AK121082) than the cDNA region (+311 bp to +825 bp) used to generate the *PUB44-kd* cell lines. We transiently expressed PBI1-SmBiT, OsCERK1 and T7:LgBiT-tagged PUB44 in rice protoplast (Fig. 2e and Supplementary Fig. 5a). The PBI1 protein levels were reduced by treatment with chitin. Co-transfection of the *PUB44 RNAi-ver2* construct reduced the protein level of PUB44, which compromised chitin-induced reduction of PBI1. These data suggest that PUB44 is involved in the chitin-induced degradation of PBI1.

The PUB44-dependent degradation of PBI1 implies that ubiquitination of PBI1 occurs during the chitin response. To test whether PBI1 is ubiquitinated by treatment with chitin, we analyzed in vivo ubiquitination of PBI1 using co-immunoprecipitation with

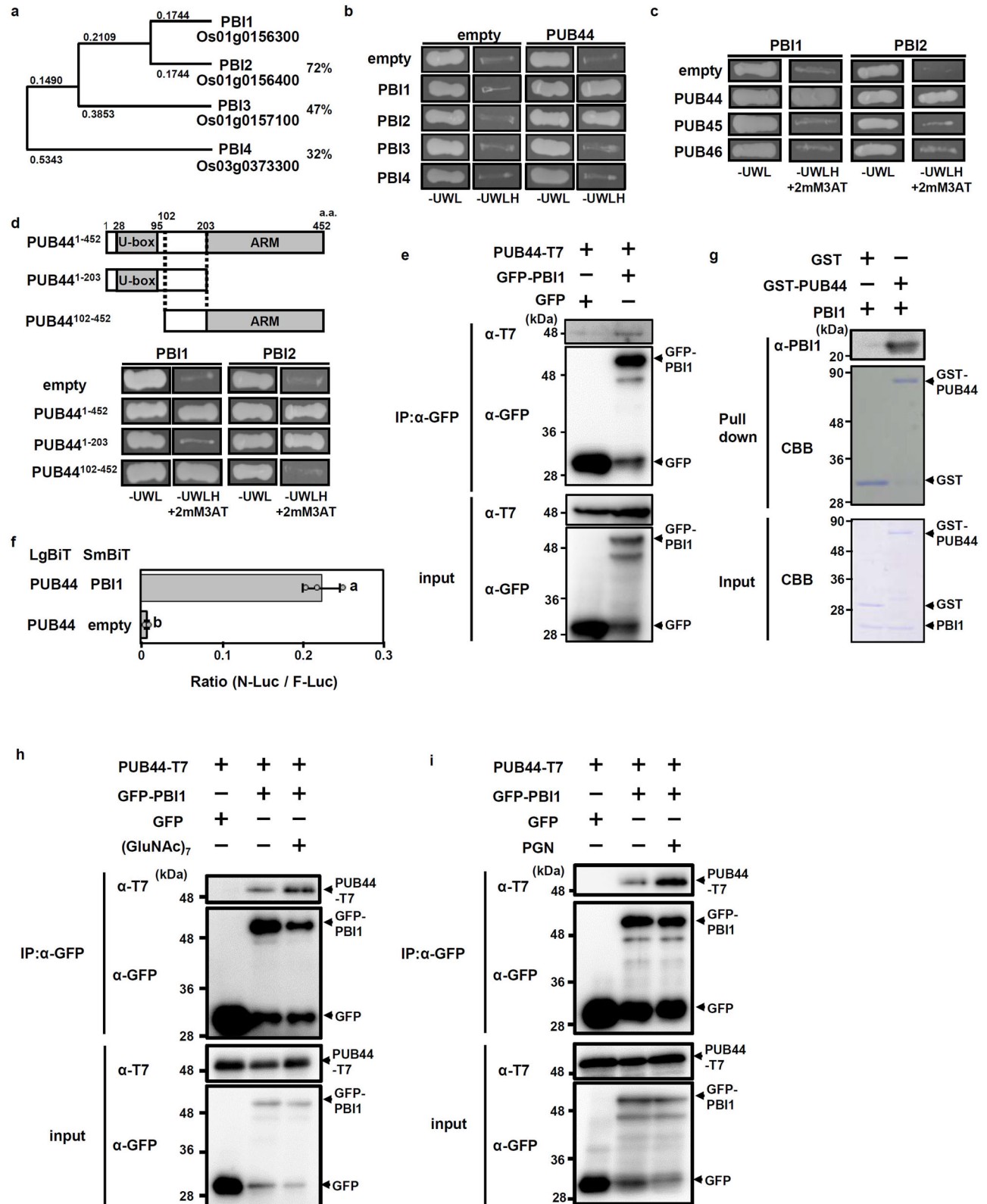

α-Ubiquitin (Fig. 2f). We found that the high-molecular weight bands that correspond to poly-ubiquitinated PBI1 protein were induced in wild type but not the *pbi1-1* mutant (the *pbi1-1* mutant was described below) by treatment with chitin. Instead, the level of unmodified PBI1 protein was reduced as consistent with Fig. 2a (input of Fig. 2f). These results strongly suggest chitin-induced ubiquitination of PBI1.

To test whether PUB44 is involved in ubiquitination of PBI1, we used the *PUB44-kd* cell line for co-immunoprecipitation assay with α-Ubiquitin. Consistent with Fig. 2d, the PBI1 protein level was not reduced by chitin treatment in the *PUB44-kd* cell (input of Fig. 2g), whereas chitin-induced degradation of PBI1 was observed in wild type. In addition, the increase of the high-molecular weight bands of PBI1 was not observed in the *PUB44-*

**Fig. 1 Proteins containing the DUF1110 domain form a small protein family in rice. a** Phylogenetic tree of the rice DUF1110 domain-containing proteins. Full-length protein sequences were used in the alignment. The neighbor-joining phylogenetic tree was created using ClustalW on the DNA Data Bank of Japan website (http://www.ddbj.nig.ac.jp/). The tree was generated from the modified alignment using Treeview X software. The percentages indicate amino acid identity with PBI1. **b** Interactions between PBI family members and PUB44 in yeast two-hybrid experiments. Growth of yeast colonies on −ULWH plates (lacking uracil, leucine, tryptophan, and histidine) indicates a positive interaction. **c** Interactions of PBI1 and PBI2 with PUB44, PUB45, and PUB46 in yeast two-hybrid experiments. Growth of yeast colonies on −ULWH plates with 2 mM 3-aminotriazole (3-AT) indicates a positive interaction. **d** (Upper panel) Schematic diagram of PUB44 constructs. (Bottom panel) Interactions of PBI1 and PBI2 with each domain of PUB44 in yeast two-hybrid experiments. Positive interactions are as for (**c**). **e** Rice protoplasts were co-transfected with GFP-PBI1 and T7-tagged PUB44 and subjected to co-immunoprecipitation assay. Proteins were precipitated using α-GFP. The input proteins and the precipitated proteins were subjected to immunoblotting with α-T7 or α-GFP. **f** Interaction between PUB44 and PBI1 was analyzed using split NanoLuc luciferase complementation assays. PUB44 and PBI1 were fused to LgBiT and SmBiT, respectively. Firefly luciferase (F-Luc) was used as an internal control. Rice protoplasts were transfected with these constructs, and the interactions were indicated by the N-Luc to F-Luc ratios. Data are means ± SD. $n = 3$ biologically independent replicates. Different letters above the data points indicate significant differences (two-sided Student's $t$-test, $p < 0.05$). **g** The in vitro pull down assay was carried out using GST-PUB44 and PBI1. The PBI1 protein was detected by immunoblotting with α-PBI1. **h** Rice protoplasts co-transfected with GFP-PBI1 and T7-tagged PUB44 were treated with or without 2 μg/ml (GluNAc)$_7$ for 10 min and subjected to co-immunoprecipitation assay. Proteins were precipitated using α-GFP. The input proteins and the precipitated proteins were subjected to immunoblotting with α-T7 or α-GFP. **i** Rice protoplasts co-transfected with GFP-PBI1 and T7-tagged PUB44 were treated with or without 100 μg/ml peptidoglycan for 10 min and subjected to co-immunoprecipitation assay. Proteins were precipitated using α-GFP. The input proteins and the precipitated proteins were subjected to immunoblotting with α-T7 or α-GFP. The experiments in (**b**–**i**) were repeated three times with similar results.

*kd* cell (Fig. 2g), indicating that PUB44 regulates PBI1 ubiquitination. Since the ligase activity of recombinant full length PUB44 protein is very weak[40], we failed to detect ubiquitination of PBI1 by PUB44 in multiple in vitro ubiquitination assays. However, the co-immunoprecipitation assay of ubiquitinated proteins suggests that PUB44 may ubiquitinate PBI1.

The *Xoo* XopP effector inhibits PUB44 activity by interacting with its U-box domain[40], and over-expression of *XopP* in plant cells suppresses pattern-triggered immunity mediated by PUB44[40]. We treated rice suspension cells that were over-expressing *XopP* (*XopP-ox* cells) with chitin and examined the levels of PBI1 protein. Chitin-induced degradation of PBI1 was significantly suppressed in the *XopP-ox* cells (Fig. 2h). In addition, transient expression of *XopP-Myc* partially inhibited chitin-induced PBI1 degradation in rice protoplast (Fig. 2i and Supplementary Fig. 5b). These data support the possibility that PBI1 degradation may be regulated by the PUB44-mediated ubiquitination pathway.

**PBI1 is composed of a four-helix bundle**. Although PBI1 consists mainly of the DUF1110 domain, the molecular nature of this domain was unknown. Therefore, we determined its tertiary structure of PBI1. The crystal structure was solved at a resolution of 1.84 Å. PBI1 is composed of a four-helix bundle (Fig. 3a, b) with a diameter of approximately 19 Å and a length of about 70 Å. There are six molecules in the asymmetric unit. The root mean square differences (r.m.s.d.) between each monomer are from 0.15 to 0.85 Å. We observed no large conformational change induced by crystal packing. The calculated solvent content was 63% (Matthews coefficient = 3.32 Å³ Da⁻¹). The four helices of PBI1 are arranged in an up-down-up-down topology, and the bundle is leftward turning. The four helices are part of a single polypeptide chain (Ala10–His39, Glu49–Gly90, Leu110–Asp148, and Val110–Val191) and are connected to each other by three loops (Leu40–Asp48, Gly91–Tyr109, and His149–Cys154). The interfaces between the helices consist of hydrophobic residues, whereas hydrophilic residues are exposed on the surfaces that interact with the aqueous environment. The hydrophobic residues occur as repeats of 3 or 4 residues per helical turn and form the core of the bundle structure.

We used the Dali server[43] to perform a database search for three-dimensional structures that exhibit similarity to PBI1, and identified eight unique proteins with Z-scores higher than 10. Seven of these proteins are: Methyl-accepting chemotaxis transducer[44], SH2 domain[45], Talin 1[46], surface protein VSPA[47], focal adhesion kinase 1[48], tyrosine kinase 2 beta[49], and super-oxide dismutase[50]. In addition, the four-helix bundle structure occurs in the CC domains of plant CC-NB-LRRs including Rx and MLA10[51–53]. In particular, the CC domain of Rx is very similar in structure to PBI1, with a high Z-score of 4.5. A structural alignment of the Rx CC domain and PBI1 showed a high degree of similarity (Supplementary Fig. 6).

**PBI1 interacts with WRKY45 in the nucleus**. To analyze the subcellular localization of PBI1, we made constructs encoding GFP fused to the N- or C-terminal of PBI1. These constructs, along with one encoding red fluorescent protein (RFP) containing a nuclear localization signal (RFP-nls), were used to transfect rice protoplasts. Fluorescence from both the GFP-PBI1 and PBI1-GFP proteins was detected in nuclei and cytoplasm (Fig. 3c).

The presence of PBI1 in the nucleus suggests that it may be involved in transcriptional regulation. To screen for rice factors that interact with PBI1, we analyzed interaction of PBI1 with some rice transcription factors reported to be involved in rice immunity using bimolecular fluorescence complementation (BiFC) assays, and identified WRKY45 as a candidate. WRKY45 is a key regulator of rice immunity against rice blast and bacterial blight diseases[24]. To analyze the interaction between PBI1 and WRKY45, we performed a BiFC assay using rice protoplasts. WRKY45 was tagged with the N-terminal domain of the yellow fluorescent protein Venus (WRKY45-Vn), and PBI1 was tagged with the C-terminal domain of Venus (PBI1-Vc). Transient expression of these constructs together resulted in fluorescence in the nucleus (Fig. 4a). We also examined the interaction between PBI1 and WRKY45 in a co-immunoprecipitation assay using rice protoplasts transiently expressing GFP-PBI1 and Myc-tagged WRKY45. Myc-WRKY45 co-immunoprecipitated with GFP-PBI1 (Fig. 4b), confirming the in vivo interaction between PBI1 and WRKY45.

Nuclear interaction between PBI1 and WRKY45 suggests nuclear localization of PUB44. To examine the possibility, PUB44-GFP was transiently expressed in rice protoplast, and analyzed by optical sectioning using a fluorescence microscope with Apotome2 system (Carl Zeiss). This result indicated that PUB44-GFP localized to both nucleus and cytoplasm (Supplementary Fig.7), suggesting that PUB44 might interact with PBI1 and WRKY45 in the nucleus.

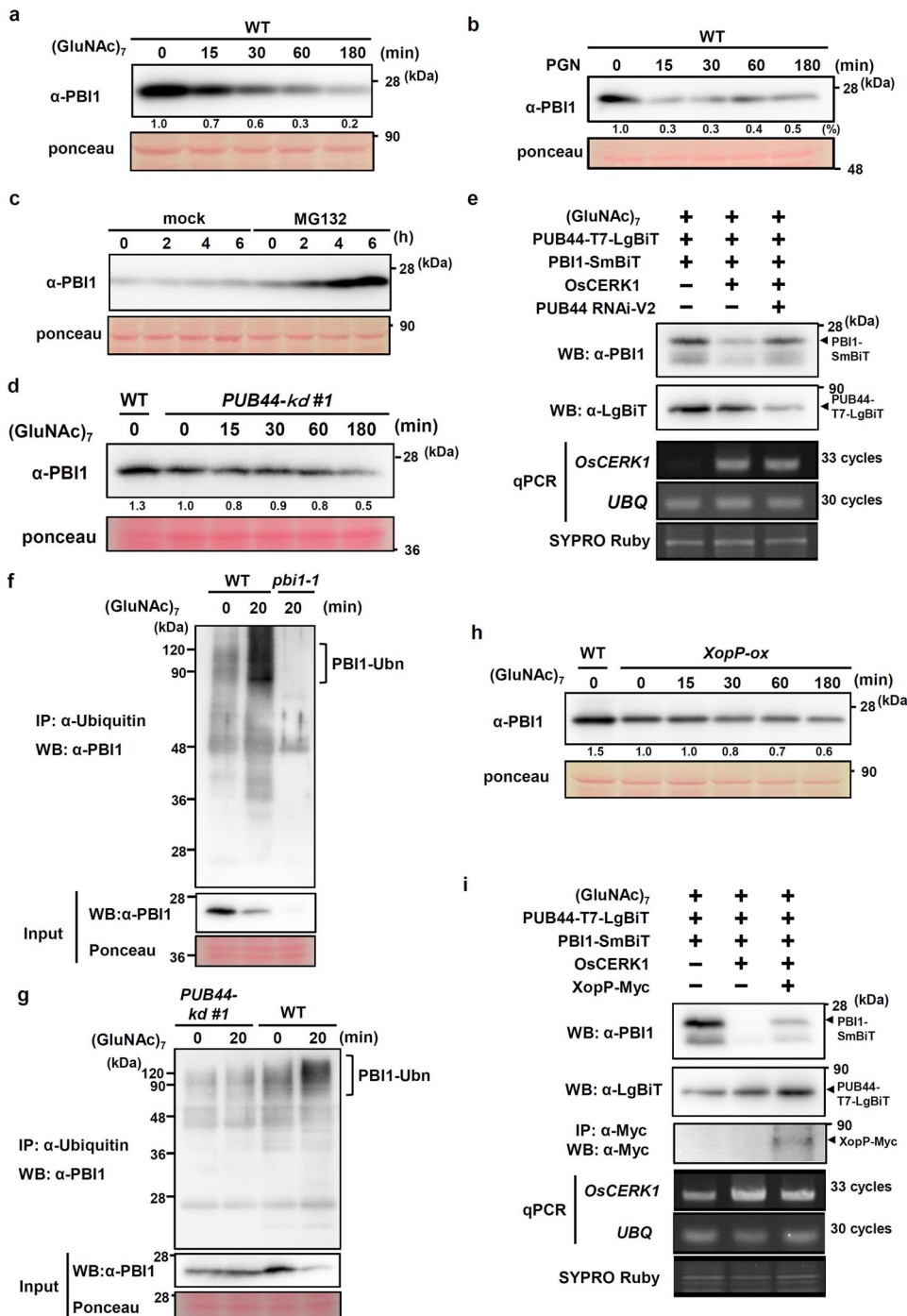

To examine whether PBI1 directly interacts with WRKY45, we carried out the in vitro pull down assay using GST-fused PBI1 and HA-tagged WRKY45. HA-WRKY45 was pulled down with GST-PBI1 but not GST (Fig. 4c), indicating direct interaction between PBI1 and WRKY45. To analyze specificity of the interaction between PBI1 and WRKY45, we tested whether PBI1 interacts with other WRKYs such as WRKY13, WRKY47, and WRKY62. However, no interaction of PBI1 with these three WRKYs was observed by the in vitro pull down assay (Supplementary Fig. 8a). Furthermore, we examined the interaction of PBI1 with these WRKY proteins using a split NanoLuc luciferase complementation assay. The luciferase activities supported the results of the in vitro pull down assay, in which PBI1 interacted with WRKY45 but not others (Supplementary

Fig. 8b). These data imply that the interaction between PBI1 and WRKY45 might be specific.

Since the level of WRKY45 protein is regulated by the ubiquitin-proteasome system[18], we examined the interaction between PUB44 and WRKY45. The interaction between PUB44 and WRKY45 was observed by the in vitro pull down assay (Supplementary Fig. 9a). However, the interaction between PUB44 and WRKY45 was very weak, because the interaction was undetectable when the column was washed with the buffer containing 0.1 % Triton-X. In addition, the split NanoLuc luciferase complementation assay indicated that the interaction between PUB44 and WRKY45 was much weaker than the interaction between PUB44 and PBI1 (Supplementary Fig. 9b). Thus, it does not seem that PUB44 is involved in the WRKY45 ubiquitination.

**Fig. 2 Chitin-induced degradation of PBI1. a** Total proteins were prepared from wild type (WT) rice suspension-cultured cells after treatment with 2 µg/ml (GluNAc)$_7$ and subjected to immunoblots with α-PBI1. The relative abundance of the PBI1 proteins detected is labeled under the blot with α-PBI1. **b** Total proteins were prepared from rice suspension-cultured cells after treatment with 100 µg/ml peptidogylcan and subjected to immunoblots with α-PBI1. The relative abundance of the PBI1 proteins detected is labeled under the blot with α-PBI1. **c** PBI1 protein levels in rice cells after treatment with the proteasome inhibitor MG132 (30 µM) or dimethylsulphoxide (DMSO; mock), determined by immunoblot analysis with α-PBI1. **d** PBI1 protein levels in *PUB44-kd* #1 cell line after treatment with 2 µg/ml (GluNAc)$_7$, determined by immunoblot analysis with α-PBI1. The relative abundance of the PBI1 proteins detected is labeled under the blot with α-PBI1. **e** Chitin-induced degradation of PBI1 was analyzed using rice protoplasts carrying the *OsCERK1, PUB44-LgBiT, PBI1-SmBiT,* and/or *PUB44 RNAi-ver2* constructs. The protoplasts treated with 2 µg/ml (GluNAc)$_7$ were subjected to immunoblotting with α-LgBiT or α-PBI1, semi quantitative RT-PCR, and quantitative real-time PCR (Supplementary Fig. 5a). **f** Ubiquitinated PBI1 proteins were analyzed using wild type and *pbi1-1* cells treated with 2 µg/ml (GluNAc)$_7$. Ubiquitinated proteins were immunoprecipitated with α-Ubiquitin. The co-immunoprecipitated proteins were subjected to immunoblotting with α-PBI1. **g** Ubiquitinated PBI1 proteins were analyzed using the *PUB44-kd* #1 cell treated with 2 µg/ml (GluNAc)$_7$. Ubiquitinated proteins were immunoprecipitated with α-Ubiquitin. The co-immunoprecipitated proteins were subjected to immunoblotting with α-PBI1. **h** PBI1 protein levels in *XopP*-ox cells after treatment with 2 µg/ml (GluNAc)$_7$, determined by immunoblot analysis with α-PBI1. The relative abundance of the PBI1 proteins detected is labeled under the blot with α-PBI1. **i** Chitin-induced degradation of PBI1 was analyzed using rice protoplasts carrying the *OsCERK1, PUB44-LgBiT, PBI1-SmBiT,* and/or *XopP-Myc* constructs. The protoplasts treated with 2 µg/ml (GluNAc)$_7$ were subjected to immunoblotting with α-LgBiT, α-PBI1 or α-Myc, semi quantitative RT-PCR, and quantitative real-time PCR (Supplementary Fig.5b). The experiments in (**a**), (**b**), (**d**), and (**h**) were repeated more than five times, and (**c**), (**e**), (**f**), (**g**), and (**i**) three times with similar results.

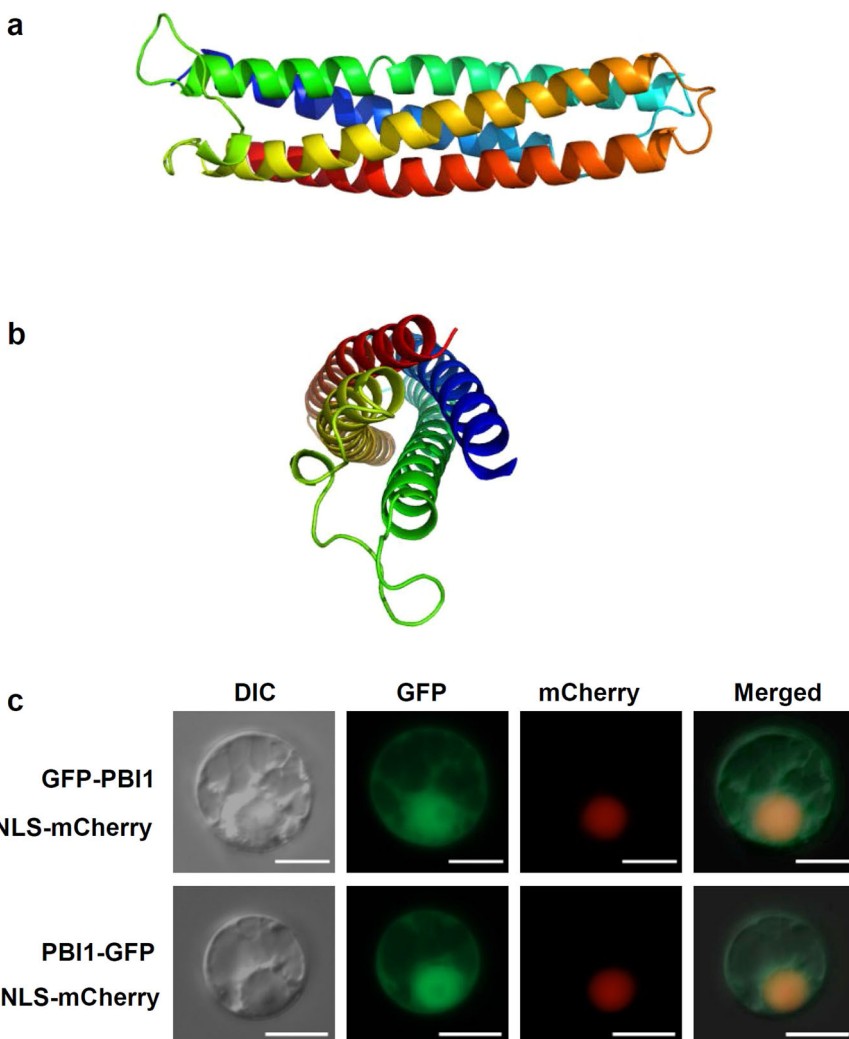

**Fig. 3 PBI1, with a four-helix bundle structure, localizes mainly to the nucleus. a** Side view of PBI1, which forms a four-helix bundle. Coloring is from blue at the N-terminus to red at the C-terminus. **b** End view, with N- and C-termini at the front. **c** Detection of GFP-PBI1 and PBI1-GFP after transient expression in rice protoplasts. mCherry with a nuclear localization signal was used as a nuclear localization marker. Scale bar = 10 µm. Similar results were obtained in three independent experiments.

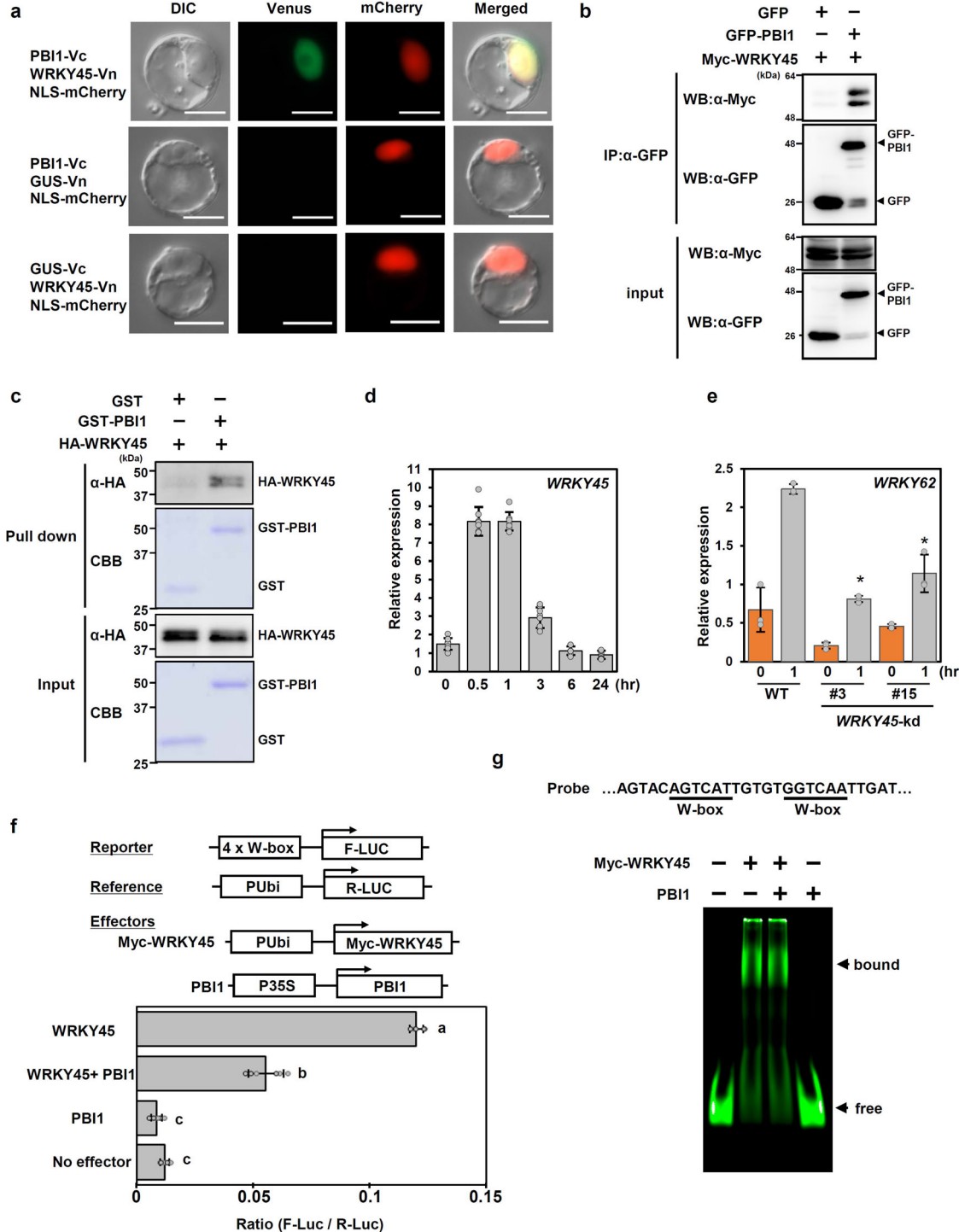

The fact that WRKY45 interacts with PBI1 raises the possibility that WRKY45 is involved in chitin-induced immunity. Quantitative real-time PCR experiments demonstrated that expression of *WRKY45* is activated after treatment with chitin (Fig. 4d). We also found that chitin-induced expression of *WRKY62*, which functions downstream of *WRKY45*[54], was significantly suppressed in two *WRKY45*-knockdown lines (Fig. 4e and Supplementary Fig. 10a)[54], suggesting the involvement of WRKY45 in chitin-induced immunity.

The interaction between PBI1 and WRKY45 in nuclei suggests that PBI1 may be involved in the regulation of transcription by WRKY45. We carried out transactivation assays using effector

constructs expressing Myc-tagged WRKY45 and PBI1, and a reporter construct containing a promoter with four W-box sequences upstream of the *luciferase* cDNA[24]. We transfected rice protoplasts with the Myc-tagged WRKY45 construct and the reporter, with or without the PBI1 construct, and examined the luciferase activity. Luciferase activity was increased in the presence of WRKY45 (Fig. 4f and Supplementary Fig. 10b) but was significantly inhibited by co-expression of PBI1. This result indicates that PBI1 negatively regulates the transcriptional activity of WRKY45.

Since PBI1 inhibited the transcriptional activity of WRKY45, it is possible that PBI1 may suppress the DNA-binding activity of

**Fig. 4 PBI1 interacts with and inhibits WRKY45. a** Bimolecular fluorescence complementation (BiFC) analysis was used to visualize the interaction between PBI1-Vc and WRKY45-Vn in rice protoplasts. mCherry with a nuclear localization signal was used as a nuclear localization marker. The β-glucuronidase (GUS) protein was used as a negative control. Scale bar = 10 µm. **b** Rice protoplasts were co-transfected with GFP-PBI1 and Myc-tagged WRKY45 and subjected to a co-immunoprecipitation assay. Proteins were precipitated using an antibody against GFP (α-GFP), and the input proteins and precipitated proteins were probed with α-Myc and α-GFP. **c** The in vitro pull down assay was carried out using GST-PBI1 and HA-WRKY45. The HA-WRKY45 protein was detected by immunoblotting with α-HA. **d** *WRKY45* transcript levels in rice suspension-cultured cells treated with 2 µg (GluNAc)$_7$ were analyzed using quantitative real-time PCR. Data are means ± SD. $n = 8$ biologically independent samples. **e** Expression levels of *WRKY62* in wild type and *WRKY45*-knockdown (kd) leaves after treatment with 2 µg/ml (GluNAc)$_7$, analyzed using quantitative real-time PCR. Data are means ± SD. $n = 3$ biologically independent replicates. The asterisks indicate statistically significant differences between the wild-type and *WRKY45*-kd leaves by the two-sided Student's *t*-test ($P < 0.05$). **f** Transactivation assay using a dual-luciferase system. The reporter construct contained four W-box sequences upstream of the F-Luc coding sequence. The Myc-WRKY45 construct contained a Myc-tagged full length *WRKY45*-coding sequence downstream of the maize *ubiquitin* promoter (pUbi). The PBI1 construct contained the *PBI1*-coding region downstream of the cauliflower mosaic virus 35 S promoter. The reference construct contained the Renilla *luciferase* (R-Luc) coding sequence downstream of the maize *ubiquitin* promoter. Luciferase activities were normalized against the reference R-Luc activity. Values are mean ± SE. $n = 6$ biologically independent replicates. Different letters above the data points indicate significant differences ($p < 0.01$, two-sided Welch's *t* test). **g** Electrophoresis mobility shift assay was performed using Myc-WRKY45 and PBI1. The probe containing two W-box sequences was labeled with IRDye800. Myc-WRKY45 was prepared using wheat germ in vitro protein synthesis system. The protein level of Myc-WRKY45 used in this experiment was below the detection limit of CBB staining. 7.8 µg of PBI1 protein was used in this assay. All the above experiments were performed and analyzed three times with similar results.

WRKY45. WRKY45 has been reported to bind to two W-box sequences in the promoter of rice *DFA* gene encoding a basic helix-loop-helix-type transcription factor[54]. We carried out an electrophoresis mobility shift assay (EMSA) using this promoter fragment as a probe. WRKY45 bound to the W-box sequences (Fig. 4g), which was suppressed by adding unlabeled probe (Supplementary Fig.10c). However, excess amount of PBI1 did not inhibit the DNA-binding activity of WRKY45 (Fig. 4g). Since the shift of bound protein complex was not detected by adding PBI1, it does not seem that PBI1 may interact with the W-box-bound WRKY45. Thus, it is unlikely that PBI1 inhibits the DNA-binding activity of WRKY45. Therefore, the molecular mechanism of how PBI1 suppresses the transcriptional activity of WRKY45 remains to be understood.

**pbi1 mutations cause increases in the protein levels of WRKY45.** To clarify the roles of PBI1 in rice immunity, we generated two *PBI1* knock-out mutants (*pbi1-1* and *pbi1-2*) using the CRISPR/Cas9 system. *pbi1-1* has a frameshift mutation caused by a 2-bp deletion, and *pbi1-2* has a 6-bp deletion causing the loss of two amino acid residues and a 1-bp substitution within the *PBI1*-coding region (Supplementary Fig. 11a). Immunoblots with α-PBI1 showed that no PBI1 protein was detected in either mutant (Fig. 5a). Since no truncated PBI1-2 protein was detected in *pbi1-2*, the deletion of the two amino acid residues may have caused protein stability.

Both *pbi1-1* and *pbi1-2* exhibited a weak dwarf phenotype (Fig. 5b), which was similar to the phenotype of *WRKY45*-oxerexpressing plants[24]. Therefore, we analyzed the protein levels of WRKY45 by immunoblotting with α-WRKY45. The WRKY45 protein levels were significantly increased in the *pbi1* mutants compared with wild type (Fig. 5c). The *WRKY45* transcript levels were also increased in the *pbi1* mutants (Fig. 5d). Since WRKY45 is known to be autoregulated[23], these results suggest that loss of PBI1 may result in the leaky autoactivation of *WRKY45* transcription. Although these data suggest that autoimmunity may occur in the *pbi1* mutants, these mutants did not exhibit cell death phenotype and upregulation of defense genes. We only detected enhanced expression of *PR10* gene in the mutants (Supplementary Fig. 11b). Since WRKY45 is known to regulate immune priming[55], it is possible that the *pbi1* mutation may induce immune priming through increase of the WRKY45 protein levels.

We examined the resistance of the *pbi1* mutants to the compatible race *Xoo* T7174 by inoculating the plants using the clipping method. The *pbi1* mutants developed disease lesions that were shorter than those of the wild type (Fig. 5e, f). Genomic quantitative PCR using specific primers for the *X. oryzae XopA* gene indicated that bacterial growth was also reduced in the *pbi1* mutants (Fig. 5g). Thus, the *pbi1* mutants enhanced resistance to *X. oryzae*, possibly via accumulation of WRKY45. In addition, we analyzed the resistance of the *pbi1* mutants to the compatible race *Magnaporthe oryzae* ken53-33. However, no significant increase of the blast resistance was observed (Supplementary Fig. 10c).

**Chitin-induced MAPK activation positively regulates PBI1 degradation.** Previously we reported that chitin perception triggers the phosphor-signaling pathway OsCERK1 – OsRLCK185 – MAPKKK11/MAPKKK18 – MKK4/MKK5 – MPK3/ MPK6[15]. To ask whether this MAPK pathway affects chitin-induced degradation of PBI1, we produced *mapkkk11/mapkkk18* double mutant lines by using the CRISPR-CAS9 system and the *mapkkk11-1* mutant background, which was generated by a *Tos17* insertion in *MAPKKK11* gene[15]. The *mapkkk11-1/mapkkk18-1* and *mapkkk11-1/mapkkk18-2* lines carry nonsense mutations caused by 1 bp insertions in *MAPKKK18* gene (Supplementary Fig. 12a). Chitin-induced activation of MPK3 and MPK6 was significantly reduced in the mutants (Fig. 6a). As shown in Fig. 6b, chitin-induced PBI1 degradation was suppressed in the *mapkkk11/mapkkk18* mutants, whereas the expression pattern of *PBI1* gene in the *mapkkk11/mapkkk18* mutant was same as that in wild type (Supplementary Fig. 12b). These data indicate that MAPK activity is required for the PBI1 degradation.

WRKY-type transcription factors are known to be activated by MAPK-mediated phosphorylation[20]. The three amino acid residues Thr266, Ser294, and Ser299, located in the C-terminal region of WRKY45, are known to be phosphorylated by MPK6 in vitro[19]. The phosphorylation of Ser294 and Ser299 positively regulates the immune response, whereas a phosphor-mimic mutation of Thr266 inhibits immunity[56]. To examine whether the phosphorylation of WRKY45 by MAPKs may affect the interaction between WRKY45 and PBI1, we analyzed the interaction using a split NanoLuc-luciferase assay. For this assay, the full length, N-terminal (1–174 aa) or C-terminal (175–326 aa) regions of WRKY45 were fused to SmBiT, and PBI1 was fused to LgBiT. The experiments indicated that PBI1 interacts more strongly with the C-terminal region of WRKY45 than with the N-terminal region (Fig. 6c). We produced a phosphor-mimic mutant (WRKY45$^{DD}$) of WRKY45 in which Ser294 and Ser299 were each substituted with Asp. The phosphor-mimic mutation

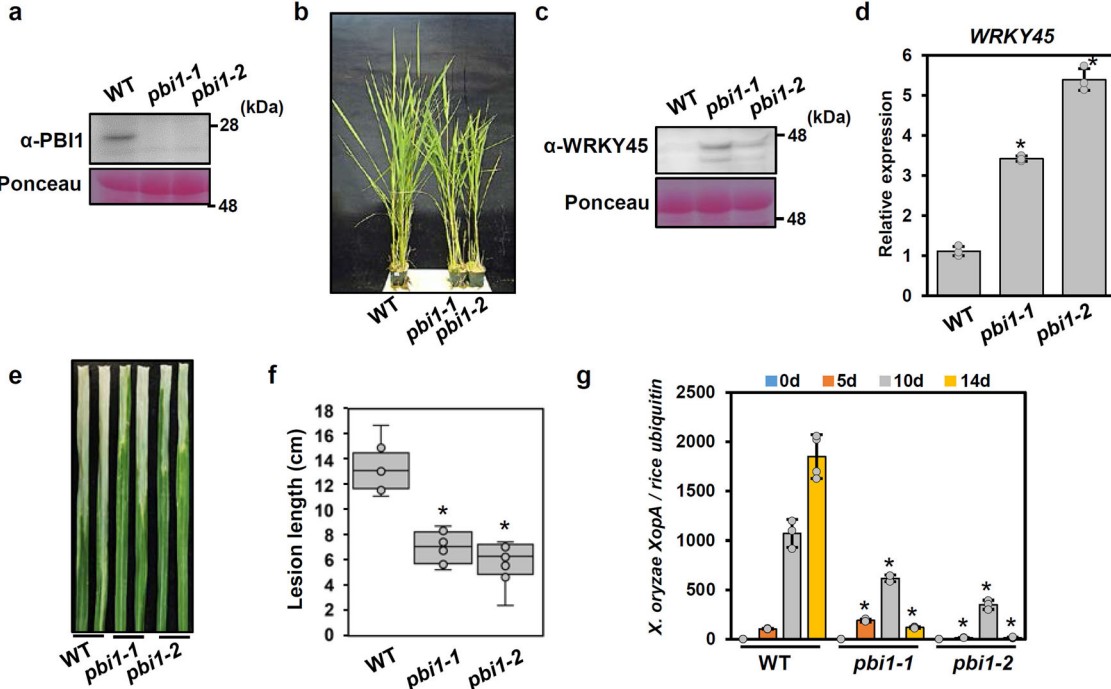

**Fig. 5 PBI1 negatively regulates disease resistance through WRKY45. a** PBI1 protein levels in leaves of the *pbi1-knockout* (*ko*) mutants *pbi1-1* and *pbi1-2* were analyzed by immunoblotting with α-PBI. **b** Phenotypes of the *pbi1-ko* mutants. **c** WRKY45 protein levels in leaves of the *pbi1-ko* mutants, analyzed by immunoblotting with α-WRKY45. **d** *WRKY45* transcripts level in leaves of the *pbi1* mutants, analyzed by quantitative real-time PCR. Error bars indicates ±SD. *n* = 3 biologically independent replicates. Asterisks indicate significant differences between the WT and the *pbi1* mutants (two-sided Student's *t*-test *P* < 0.01). **e** Rice leaves were infected with *Xoo* T7174 using a crimping method. The photograph of disease lesions was taken at 14 dpi. Scale bar = 1 cm. **f** Mean lengths of disease lesions at 25 dpi. The box plots show the first and third quartiles as bounds of box, split by the medians (lines), with whiskers extending 1.5-fold interquartile range beyond the box, and minima and maxima as error bar. Error bars indicate ±SD. *n* = 8 biologically independent samples. Asterisks indicate significant differences between the WT and the *pbi1* mutants (two-sided Student's *t*-test *P* < 0.01). **g** The bacterial populations of *Xoo* T7174 were analyzed by quantitative real-time PCR. The data indicate the DNA levels of the *X. oryzae XopA* gene relative to that of the rice *ubiquitin* gene. Error bars indicate ±SD. *n* = 4 biologically independent replicates. Asterisks indicate significant differences between the WT and the *pbi1* mutants (two-sided Student's *t*-test *P* < 0.01). The experiments in (**a**) and (**c–g**) were repeated three times with similar results.

suppressed the interaction between WRKY45 and PBI1 (Fig. 6d). In addition, we tested the effect of the dominant-active phosphor-mimic mutant (MKK4$^{DD}$) of MKK4, because expression of MKK4$^{DD}$ induces activation of MPK3 and MPK6[57]. The interaction between PBI1 and WRKY45 was also suppressed by co-expression with MKK4$^{DD}$ (Fig. 6d). Our data indicate that the phosphorylation of WRKY45 may reduce the binding affinity between PBI1 and WRKY45. It seems that this reduced binding affinity might stimulate the PUB44-mediated degradation of PBI1. In fact, chitin-induced expression of *WRKY62* was strongly reduced in the *mapkkk11/mapkkk18* mutants (Fig. 6e), whereas expression of *WRKY45* was less affected by these mutations. These results suggest that the MAPK-mediated phosphorylation of WRKY45 and the PUB44-mediated degradation of PBI1 function cooperatively in the activation of WRKY45.

During these experimental processes, we found a shifted band of PUB44 on the immunoblots as shown by an arrow in Fig. 7a. The shifted band of PUB44 was detected by treatment with chitin. However, the shifted band was undetectable in the *Oscerk1*[11], and it disappeared after treatment with λ phosphatase (Fig. 7b). These results indicate that PUB44 is phosphorylated in an OsCERK1-dependent manner. The phosphorylation of PUB44 was delayed and reduced in the *mapkkk11/mapkkk18* mutant (Fig. 7c). In addition, the transcript level of *OsCERK1* in the *mapkkk11/18* mutant was lower than in wild type cells (Fig. 7d), suggesting that the steady-state levels of *OsCERK1* transcript are controlled through the MAPK pathway. The reduced levels of *OsCERK1* transcript in the *mapkkk11/18* mutant may be the reason for the

reduction and delay in PUB44 phosphorylation. Thus, it is possible that the defects in PBI1 degradation in the *mapkkk11/mapkkk18* mutants are partially associated with the reduction of PUB44 phosphorylation which may be caused by reduced expression of *OsCERK1*.

## Discussion

PUB44 was originally identified as the target for *X. oryzae* type III effector XopP. Previous study indicated that PUB44 plays an important role in immune activation in response to bacterial peptidoglycan as well as fungal chitin. In rice, upon perception of peptidoglycan and chitin, the corresponding PRRs transmit the immune signals into intracellular components through OsCERK1[9–11]. Therefore, PUB44 is most likely activated downstream of OsCERK1. However, the molecular mechanisms of how PUB44 is activated and how it regulates the downstream immune responses had been unknown so far. In this study, we found that upon perception of chitin, PUB44 is phosphorylated in an OsCERK1-dependent manner. We also identified PBI1 as an interactor with PUB44. During the chitin response, PBI1 is degraded in a PUB44-dependent manner, suggesting that PUB44 may control immunity through degradation of PBI1. In addition, PBI1 interacts with and inhibits WRKY45, a key regulator of rice immunity. PBI1 degradation is also regulated by MAPKs. The data presented here demonstrate that the chitin-induced activation of WRKY45 is regulated by both MAPK-mediated phos-phorylation and PUB44-mediated PBI1 degradation.

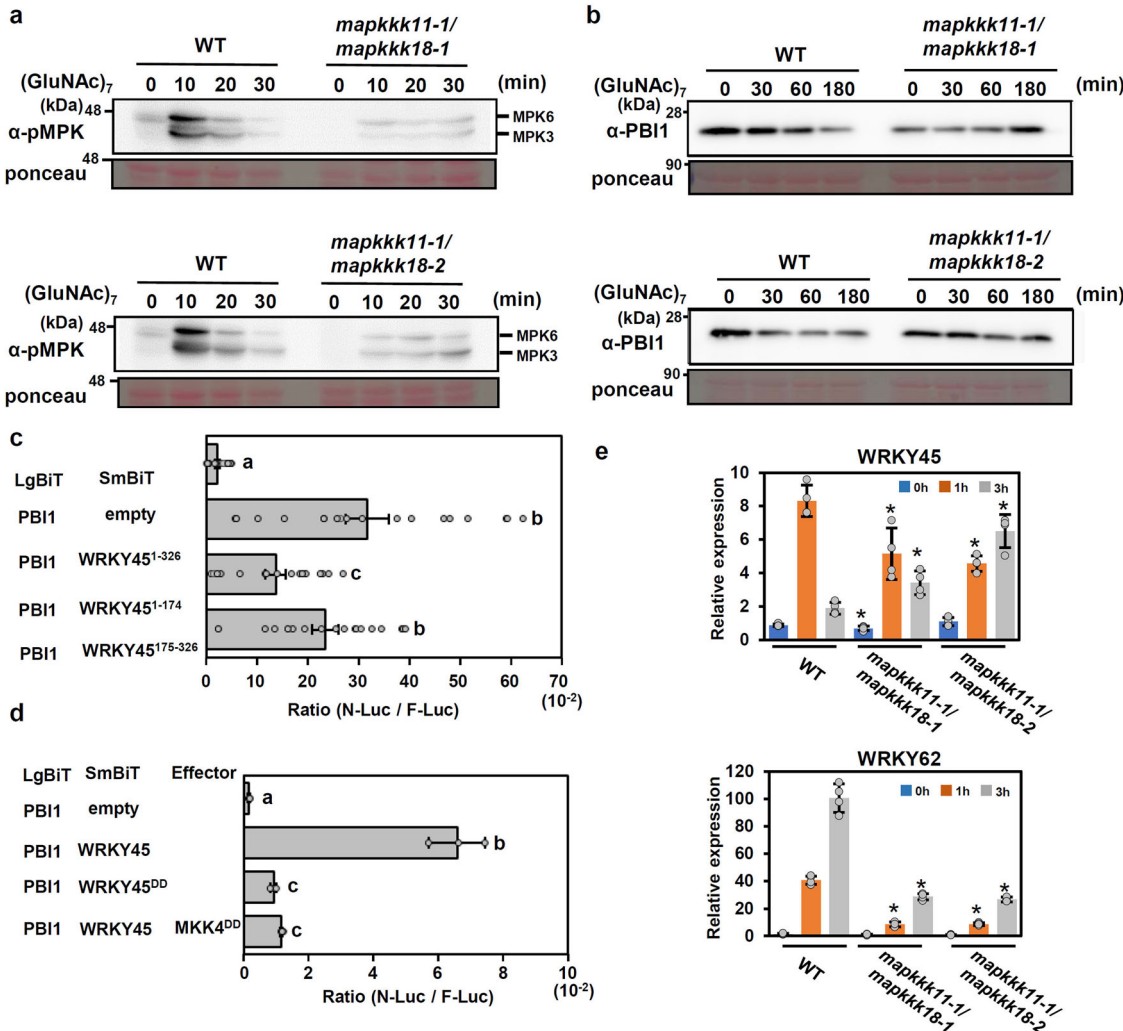

**Fig. 6 MAPKs regulate PBI1 degradation. a** Chitin-induced MAPK activation in two *mapkkk11/mapkkk18* mutants. Total proteins were prepared from rice suspension-cultured cells after treatment with 2 μg/ml (GluNAc)$_7$ and subjected to immunoblots with α-pMAPK. **b** Chitin-induced PBI1 degradation was inhibited in the *mapkkk11/mapkkk18* mutants. Total proteins were prepared as for (**a**) and probed with α-PBl1. **c** The interactions between PBI1 and full length WRKY45 or WRKY45 fragments were analyzed using split NanoLuc luciferase assays. PBI1 was fused to LgBiT, and the WRKY45 fragments were fused to SmBiT. F-Luc was used as an internal control. Rice protoplasts were transfected with the constructs and the interactions were indicated by the N-Luc to F-Luc ratios. Data are means ± SE. $n = 19$ biologically independent replicates. Different letters above the data points indicate significant differences ($p < 0.01$, two-sided Welch's *t* test). **d** Phosphor-mimic mutation of WRKY45 reduces the interaction with PBI1. Split NanoLuc luciferase assays were carried out by transient expression of PBI1-LgBiT and WRKY45-SmBiT or WRKY45$^{DD}$-SmBiT with or without MKK4$^{DD}$ in rice protoplasts. Values are means ± SE. $n = 3$ biologically independent replicates. Different letters above the data points indicate significant differences ($p < 0.01$, two-sided Welch's *t* test). **e** The expression levels of *WRKY45* and *WRKY62* in *mapkkk11/mapkkk18* suspension-cultured cells treated with 2 μg/ml (GluNAc)$_7$ were analyzed using quantitative real-time PCR. Data are means ± SD. $n = 4$ biologically independent replicates. The asterisks indicate statistically significant differences from the WT controls by two-sided Student's *t*-test ($P < 0.05$). All above experiments were repeated three times with similar results.

PBI1 is a novel protein carrying the DUF1110 domain, and it forms a small protein family with PBI2, PBI3, and PBI4. The biological function of this family has not been elucidated so far. In this study, we determined the crystal structure of PBI1 and found that it forms a four-helix bundle. Many other proteins have four-helix bundle structures, including the CC domains of the CC-NLR-type immune receptors[51,53]. In fact, the tertiary structure of PBI1 is very similar to that of the CC domain of Rx, which is a CC-NLR receptor. Interestingly, it has been reported that WRKY45 interacts with the CC domain of Pb1, a CC-NLR protein involved in rice blast resistance[22]. Pb1 is predicted to positively regulate the abundance of WRKY45 protein by protecting it from degradation by the ubiquitin proteasome system, however, the molecular mechanisms have not been elucidated in

detail. In contrast to Pb1, PBI1 appears to negatively regulate the abundance of WRKY45 protein, because WRKY45 protein levels are higher in the *pbi1* mutants than in the wild type.

The plant PUB family regulates a variety of biological responses, but the mechanisms of PUB activation remain largely unknown. In Arabidopsis, the activation of PUB22 is regulated by MAPK-mediated phosphorylation[58]. In this study, we found that PUB44 is phosphorylated in an OsCERK1-dependent manner upon chitin perception. The phosphorylation of PUB44 was also observed in the *mapkkk11/mapkkk18* mutants, but it was delayed and reduced. Therefore, it is unlikely that MAPKs phosphorylate PUB44. The reduced level of phosphorylation may be explained by the fact that *OsCERK1* expression was reduced in the *mapkkk11/mapkkk18* mutants. The identification of protein

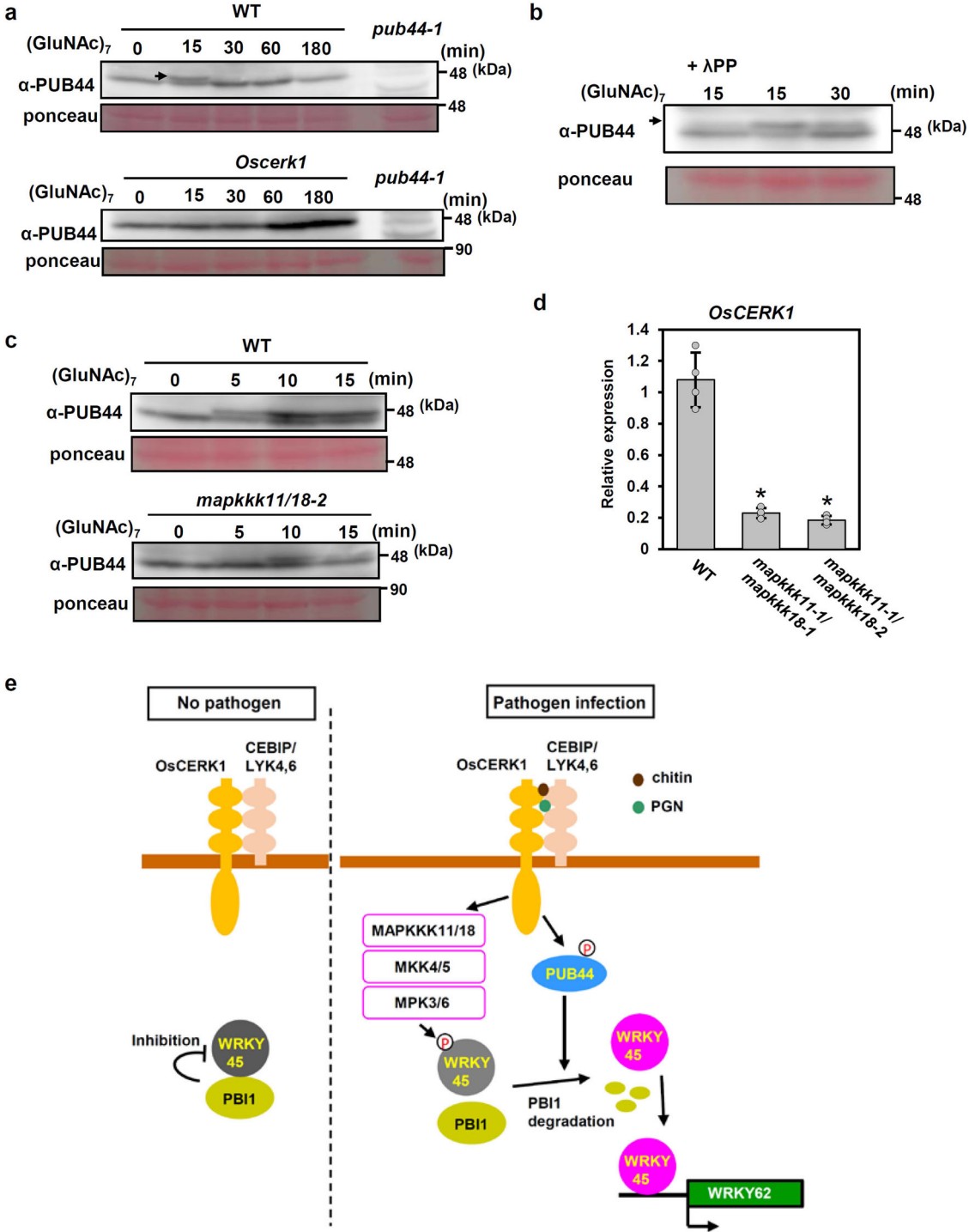

**Fig. 7 PUB44 is phosphorylated upon chitin perception. a** A mobility shift of PUB44 was detected by immunoblotting with α-PUB44 using total proteins prepared from WT rice suspension-cultured cells after treatment with 2 μg/ml $(GluNAc)_7$ (upper panel). The arrow indicates the shifted PUB44 band. The mobility shift did not occur in *Oscerk1* mutant cells (lower panel). **b** The mobility shift of PUB44 was reversed by treatment with λ protein phosphatase, indicating that the shift was due to phosphorylation of PUB44. The arrow indicates the shifted PUB44 band. **c** PUB44 phosphorylation was delayed and reduced in the *mapkkk11/18* mutant (lower panel) when compared with WT cells (upper panel). **d** *OsCERK1* transcript levels in WT and *mapkkk11/18* mutant cells, measured by quantitative real-time PCR. Data are means ± SD. $n = 4$ biologically independent replicates. The asterisks indicate statistically significant differences from the WT control by two-sided Student's *t*-test ($P < 0.05$). **e** Proposed model of WRKY45 activation. Under unelicited conditions, PBI1 inhibits WRKY45 activation in order to maintain its basal activity. Upon perception of chitin / PGN, the MAPK cascade is activated and MAPKs phosphorylate WRKY45, which reduces the binding affinity between WRKY45 and PBI1. At the same time, PUB44 is phosphorylated in OsCERK1-dependent manner, and then PBI1 is degraded. The phosphorylation of WRKY45 by MAPKs and the PBI1 degradation fully activate WRKY45. Experiments in (**a**–**d**) were performed three times with similar results.

kinases that phosphorylate PUB44 will be required for a further understanding of PUB44 activation.

The co-expression of PBI1 and WRKY45 in rice protoplasts indicated that PBI1 inhibits the transcriptional activity of WRKY45. Therefore, it is likely that PBI1 functions as a negative regulator of WRKY45 by direct interaction. In fact, the *pbi1* plants contained increased levels of WRKY45 protein, possibly because of the leaky autoactivation of *WRKY45* transcription. These increased levels of WRKY45 resulted in enhanced resistance to *Xoo*, which is consistent with previous observations that overexpression of *WRKY45* enhanced *Xoo* resistance[17]. On the other hand, the enhanced levels of *WRKY45* mRNAs negatively affect plant growth[24,25]. Therefore, it is possible that negative regulation of the WRKY45 transcriptional activity via PBI1 under unelicited condition is important for growth and reproduction.

PBI1 inhibits the activity of WRKY45. Therefore, it is possible that the chitin-induced degradation of PBI1 releases WRKY45 and activates WRKY45-mediated transcription. The WRKY45 activity is known to be regulated through the phosphorylation of its C-terminal region by MPK6[19]. In fact, ectopic activation of MPK6 increases the transcriptional activity of WRKY45[19]. Consistent with this, we found that the *mapkkk11/mapkkk18* mutations greatly reducing the activation of MPK3 and MPK6 strongly suppressed the chitin-induced expression of *WRKY62*. These results indicate that the MAPKs regulate the chitin-induced activation of WRKY45. It has been shown that the DNA-binding activity of WRKYs is regulated via MAPK-mediated phosphorylation[20], however, it hasn't yet been shown that MPK3 and MPK6 control WRKY45 activity in a similar manner.

This study and previous reports indicate that WRKY45 is regulated by both MAPK-mediated phosphorylation and PBI1 degradation. In addition, we also revealed a connection between MAPK-mediated phosphorylation and PBI1 degradation. The phosphor-mimic mutations at the Ser residues of WRKY45 phosphorylated by MAPKs reduces the binding affinity between PBI1 and WRKY45, suggesting that phosphorylation of WRKY45 may stimulate the release of WRKY45 from PBI1. Furthermore, PBI1 degradation was suppressed in the *mapkkk11/mapkkk18* mutants. Thus, it is possible that the disassociation between PBI1 and WRKY45 may be associated with the PUB44-mediated degradation of PBI1.

Our study has revealed two regulatory mechanisms for WRKY45 activation, with both positive and negative regulation (Fig. 7e). Under unelicited conditions, PBI1 inhibits WRKY45 activation in order to maintain its basal activity. Upon chitin perception, the MAPK cascade is activated, and MAPKs may phosphorylate WRKY45. At the same time, PUB44 is phosphorylated and ubiquitinates PBI1, leading to PBI1 degradation. Thus, it is possible that the combined action of the PBI1 degradation and the phosphorylation of WRKY45 by MAPKs may activate WRKY45.

The perception of microbe-associated molecular patterns induces the rapid transcription of immune-related genes, which is important for effective inhibition of pathogen growth. The protein phosphorylation- and ubiquitination-based mechanisms that control the activities of transcription factors are likely able to induce expression of downstream genes much more rapidly than mechanisms involving the transcriptional control of genes encoding transcription factors. Therefore, it seems that the cooperative regulation of WRKY45 via both the PUB44-PBI1 and MAPK-pathways contributes to the rapid activation of immunity in rice. Although WRKY45 is a key factor for the activation of rice immunity, its enhanced activation negatively affects plant growth[24,25]. Therefore, the strict regulation of WRKY45 may be required for balancing immunity and growth.

## Methods

**Plant materials**. Rice (*Oryza sativa*) *Japonicum* cultivar *Nipponbare* was used as the wild type. The *pbi1*, *pub44*, and *mapkkk18* mutants were generated using the CRISPR/Cas9 system as described below. The mutants *mapkkk11*[15], *WRKY45-kd*[23], and *PUB44-kd* and *XopP-OX*[40] were described previously.

**Plasmid constructs**. Full length cDNAs for *PBI1* (Os01g0156300), *PBI2* (Os01g0156400), *PBI3* (Os01g0157100), *PBI4* (Os03g0373300), and *WRKY45* (Os05g0322900) were amplified by PCR from cDNAs prepared from wild-type rice leaves using gene-specific primers (Supplementary Table 2) and ligated into the pENTR/D-TOPO cloning vector. For the *PUB44 RNAi-ver2* construct, the 359 bp DNA fragment corresponding to 3' region (+1,204 to +1,562) of PUB44 cDNA clone (accession AK121082) was amplified by PCR using gene-specific primers (Supplementary Table 2), and transferred using the Gateway system with LR clonase reactions into pANDA-mini vector[59]. The plasmids containing *PUB44*, *PUB45*, and *PUB46* were described previously[40]. The *PBI1*, *WRKY45*, and *PUB44* cDNA fragments were transferred using the Gateway system with an LR clonase reaction into p35S-GFP-GW for the subcellular localization assays and into p35S-Vn-GW and p35S-Vc-GW for the BiFC assays[13]. For the two hybrid assays, DNA fragments of the *PBI1–PBI4* and *PUB44–PUB46* coding regions were transferred using the Gateway system with an LR clonase reaction into vectors pBTM116 (bait) and pVP16 (prey)[60].

**Generation of knock out mutants using CRISPR/Cas9 system**. The guide RNA cloning vector pU6gRNA and the all-in-one Cas9/gRNA vector pZDgRNA_Cas9ver.2_HPT were kindly provided by Dr. Endo[61]. The 20 bp sequences from +5 to +24 of *PBI1* (CGGCGGAGGCGTGGAGATCG) and from +337 to +356 of *MAPKKK18* (ATCTCGAGGACCGCGAGTAA) were selected as the target sites of Cas9 by using the CRISPR-P website (http://cbi.hzau.edu.cn/cgi-bin/CRISPR). These were cloned into pZDgRNA_Cas9ver.2_ HPT[61], and the constructs were introduced into embryogenic WT rice calli by Agrobacterium-mediated transformation[62]. To identify the knockout mutants, genomic DNA was extracted from hygromycin-resistant calli or regenerated plants. The genomic regions containing the Cas9 target sites were amplified by PCR and sequenced as described[61].

**Yeast two-hybrid assays**. The bait and prey vectors of $PUB44^{1-452}$, $PUB44^{1-203}$, $PUB44^{102-452}$, *PUB45*, and *PUB46* were described previously[40]. The bait vector carrying $PUB44^{102-452}$ and the rice cDNA library prepared from chitin-treated suspension cells[40] were used to screen the PUB44 interactors by yeast two-hybrid assay. For construction of the bait and prey vectors, the coding regions of *PBI1*, *PBI2*, *PBI3*, and *PBI4* in the pENTR/D-TOPO cloning vector were transferred by the Gateway system using an LR clonase reaction into pBTM116-GW and pVP16-GW, respectively. The bait and prey vectors were transformed into cells of *Saccharomyces cerevisiae* L40 strain. Transformants were selected on minimal medium lacking histidine, tryptophan, and leucine with or without 2 mM 3-aminotriazole (3-AT). The interaction was analyzed based on the requirement for histidine for yeast growth.

**PAMP treatments**. Rice suspension-cultured cells were subcultured for 3 days in fresh medium, divided into 12-well plates (150 mg cells, 2 ml fresh medium per well), and treated with 2 μg/ml (GlcNAc)$_7$[5] or 100 μg/ml peptidoglycan (Sigma-Aldrich 77140).

**RNA isolation and quantitative real time PCR**. Total RNA was isolated from rice suspension-cultured cells, protoplasts and leaves using TRIzol reagent (Invitrogen) and then treated with RNase-free DNase I (Roche). First-strand cDNA was synthesized from 1 μg total RNA with an oligo-dT primer and ReverTra Ace reverse transcriptase (Toyobo). Expression levels were quantified by quantitative real time PCR using gene-specific primers and the SYBR Green master mix (Applied Biosystems) in a Step-One Plus Real-Time PCR system (Applied Biosystems). Primer sequences are listed in Supplementary Table 2. The expression levels were normalized against a *ubiquitin* reference gene. Three biological replicates were used for each experiment, and two quantitative replicates were performed for each biological replicate. Expression level of *OsCERK1* was quantified by semi qRT-PCR using *OsCERK1*-sesific primer (Supplementary Table 2). The PCR reaction using Paq5000 DNA Polymerase (Agilent) was carried out according to manufacture instruction. The PCR reaction aliquot was analyzed on 1.5% agarose gel stained with ethidium bromide. *Ubq* was used as an internal control.

**Protein extraction and immunoblotting**. Total protein was extracted in a buffer containing 100 mM Tris-HCl pH 7.5, 20% (v/v) glycerin, 1% (v/v) Triton X and a protease inhibitor cocktail (Roche), separated on polyacrylamide gels, and stained with SYPRO® Ruby gel stain (Thermo Fisher science) according to manufacture instruction. The stained gel was visualized using a UV light by E-box vx2 (ViL-BER). Total protein was also analyzed by protein immunoblotting with α-PBI1 (dilution, 1:1000), α-PUB44[40] (dilution, 1:1000), α-WRKY45[22] (dilution, 1:1000), α-LgBiT (Promega, N7100, dilution, 1:500) or α-pMAPK (Cell Signaling, #4370, dilution, 1:2000). Polyclonal antibodies against PBI1 (prepared by Medical

Biological Laboratories) were raised in rabbits using the full length PBI1 protein as the antigen. For λ phosphatase treatment, total protein was incubated with λ phosphatase (Santa Cruz Biotechnology, 200312 A) and 2 mM MnCl2 at 30 °C for 90 min and subsequently subjected to immunoblotting. For detection of ubiquitinated protein, total protein was incubated with Anti-Multi Ubiquitin mAb-Magnetic Beads (Medical Biological Laboratories) at 4 °C for 12 h. After washing the beads three times, bound proteins were eluted with 20 µl of Laemmli's sample buffer and subjected to SDS-PAGE and immunoblotting.

**In vitro pull down assay.** The GST-PBI1 and GST-PUB44 proteins were prepared as described previously[40,42]. The plasmids for purification of HA or Myc-tagged WRKY45, WRKY13, WRKY47 and WRKY62 proteins were described previously[54]. The Myc-WRKY45 protein was prepared using wheat germ in vitro protein synthesis kit (Premium PLUS Expression Kit; CellFree Sciences). For purification of HA-tagged proteins of WRKY45, WRKY13, WRKY47, and WRKY62, the DNA fragments encoding HA-tagged WRKYs were PCR-amplified using corresponding plasmids as the templates. The resultant PCR products were used for wheat germ in vitro protein synthesis (Premium PLUS Expression Kit; CellFree Sciences). The GST-fused PBI1 or PUB44 protein coupled to glutathione Sepharose 4B beads (GE Healthcare) was incubated with the WRKY proteins in buffer A (50 mM Tris pH7.4, 150 mM NaCl, 5 mM EDTA, 5% Glycerol, 0.1% Triton X-100, and protease inhibitor) for 12 h at 4 °C. After washing the beads three times with the buffer A, bound proteins were eluted with 40 mM glutathione in PBS buffer (pH8.0) and subjected to SDS-PAGE and immunoblotting.

**Transient assays using rice protoplasts.** Protoplasts were isolated from cultured rice cells by digestion of the cell walls with Cellulase RS (Yakult) and Macerozyme R-10 (Yakult) as described previously[58]. Aliquots (100 µl) of protoplasts ($2.5 \times 10^6$ cells/ml) were transformed with plasmid DNA using the polyethylene glycol (PEG) method[63]. For the localization analysis and the BiFC assays, transfected protoplasts were observed using a fluorescence microscope, the Axio Imager M2 (Carl Zeiss) with the ApoTome2 system (Carl Zeiss). The transactivation assay of WRKY45 was carried out as described previously[24]. The reporter plasmid contained the firefly (F)-LUC gene downstream of a promoter containing $4 \times$ W-box sequences[24]. The Myc-tagged WRKY45 construct with the ubiquitin promoter[22] and the p35S-PBI1 construct were used as the effectors. The Renilla (R)-LUC gene under the control of the ubiquitin promoter was used as the internal control, and transcriptional activity was measured as the ratio of LUC activities (F-LUC/R-LUC). For silencing of PUB44 in rice protoplasts, the PUB44 RNAi-ver2 plasmid was transfected into rice protoplasts. After 48 h incubation at 30 °C, the protoplasts were treated with 2 µg/ml (GlcNAc)$_7$ for 30 min.

**Co-immunoprecipitation.** To analyze the interaction between PBI1 and WRKY45, rice protoplasts transiently expressing GFP-PBI1 and Myc-tagged WRKY45 were frozen in liquid nitrogen and resuspended in extraction buffer (50 mM Tris-HCl (pH 7.5), 150 mM NaCl, 10% (v/v) glycerol, 5 mM DTT, 2.5 mM NaF, 1.5 mM Na$_3$VO$_4$, 1× Complete EDTA free protease inhibitor cocktail (Roche) and 2% (v/v) IGEPAL CA-630 (MP Biomedicals)). The supernatant was incubated with GFP-Trap beads (Chromotek) for 2 h. The beads were washed four times with the extraction buffer and resuspended in an 100 µl of 1× SDS sample buffer. Co-immunoprecipitated proteins were analyzed by immunoblots with α-GFP (Proteintech, 50430-2, dilution, 1:2000) or α-Myc (Nakarai tesque, MC045, dilution, 1:2000).

To analyze the interaction between PBI1 and PUB44, rice protoplasts transiently expressing GFP-PBI1 and T7-tagged PUB44 were incubated for 6 h at 30 °C. The protoplasts treated with (GlcNAc)$_7$ or PGN for 10 min were frozen in liquid nitrogen and resuspended in extraction buffer (100 mM Tris-HCl pH 7.5, 20% (v/v) glycerol, 1% (v/v) Triton X and a protease inhibitor cocktail (Roche)). The supernatant was incubated with GFP-Trap beads (Chromotek) for 30 min. The beads were washed four times with the wash buffer (100 mM Tris-HCl pH 7.5, 20% (v/v) glycerol and 150 mM NaCl) and resuspended in an 80 µl of 1× SDS sample buffer. Co-immunoprecipitated proteins were analyzed by immunoblots with α-GFP (Proteintech, 50430-2, dilution, 1:2000) or α-T7 (Medical Biological Laboratories, PM022, dilution, 1:2000).

For detection of the Myc-tagged XopP protein, rice protoplasts transiently expressing XopP-Myc together with PUB44-T7-LgBIT, PBI1-SmBIT, and OsCERK1 were incubated at 30 °C for 24 h. The protoplasts treated with (GlcNAc)$_7$ for 30 min were frozen in liquid nitrogen and resuspended in extraction buffer (100 mM Tris-HCl pH 7.5, 20% (v/v) glycerol, 1%(v/v) Triton X and a protease inhibitor cocktail (Roche)). The supernatant was incubated with Myc-Trap beads (Chromotek) for 2 h. The beads were washed four times with the extraction buffer without Triton X and resuspended in a 50 µl of 1× SDS sample buffer. Co-immunoprecipitated proteins were analyzed by immunoblots with α-Myc (Nakarai tesque, MC045, dilution, 1:2000).

**Electrophoresis mobility shift assay.** The WRKY45-Myc protein was prepared as described above. The PBI1 protein was prepared as described previously[42]. 7.8 µg of PBI1 protein was used in this assay. 5' end of a DNA oligonucleotide (AGTTG GCAAAAGCTAAGTACA**AGT**CATTGTGT**GGT**CAATTGATCGACTGATCT GTTGCTC) was labeled with IRDye800 (Integrated DNA Technologies). The purified proteins were incubated with 1 µmol IRDye800-labeled probe DNA and 0.15 µg Poly [d(I-C)] (Roche) in binding buffer (25 mM HEPES-KOH pH7.6, 40 mM KCl, 0.1% (v/v) Nonidet P-40, 10 µM ZnCl$_2$, 1 mg/mL bovine serum albumin, 10% glycerol, 1 mM dithiothreitol). 40-fold molar excess of non-labeled oligonucleotide duplex were used for competition reactions. After 25 min incubation at room temperature, the mixtures were loaded onto a 6% polyacrylamide gel in Tris-borate buffer and electrophoresed at 4 °C. After electrophoresis, IRDye800-labeled DNA was detected by the Odyssey CLx Imaging System (LI-COR Biosciences).

**Crystallography.** The over-expression, purification, crystallization, and preliminary X-ray analysis of native and selenomethionine-labeled PBI1 were performed as described previously[42]. The experimental phase and density modification were calculated from SAD data using SHELXC/D/E[64]. Thirty-four of 36 selenium sites were identified. After density modification, the figure of merit improved from 0.38 to 0.66. Automatic model building was performed using Buccaneer[65]. Further structure refinement was performed with Coot[66] and REFMAC5[67]. The coordinates and structure factors have been deposited with the PDB (http://pdbj.org) with accession code 7X8V. Data collection and refinement statistics are given in Supplementary Table 1. The structural model was evaluated using Rampage[68].

**Split NanoLuc Luciferase complementation assay.** DNA fragments of PBI1, WRKY45$^{1-326}$, WRKY45$^{1-174}$, WRKY45$^{175-326}$, WRKY45$^{DD}$, PUB44 were transferred using the Gateway system with LR clonase reactions into p35S-LgBiT-T7-GW or p35S-SmBiT-T7-GW[41]. The plasmid containing MKK4$^{DD}$ was described previously[13]. The Firefly Luciferase gene under the control of CaMV 35S promoter was used as an internal control. The indicated combinations of plasmids were used to transfect rice protoplasts. After 18 h incubation at 30 °C, the activities of the Firefly and NanoLuc luciferases were measured on a TriStar2 LB942 luminometer (Berthold) using the ONE-Glo Luciferase Assay System (Promega) and the Nano-Glo Live Cell Assay System (Promega).

**Pathology assays.** Fully expanded rice leaves were inoculated with a compatible race of bacterial blight pathogen Xanthomonas oryzae pv. oryzae T7174 by clipping the leaf tips with scissors that had been immersed in bacterial suspension (OD600 = 0.2). Symptoms were scored by measuring lesion length 14 days after infection. The bacterial population of Xoo T7174 was also analyzed by quantitative real-time PCR. The DNA levels of the Xoo XopA gene relative to those of the rice ubiquitin gene were measured using genomic DNAs purified from the infected leaves.

**Reporting summary.** Further information on research design is available in the Nature Research Reporting Summary linked to this article.

## Data availability
The authors declare that all data supporting the findings of this study are available within this article and its Supplementary Information files. The atomic coordinates and the structure factor files have been deposited in the Protein Data Bank (PDB) under accession number 7X8V. Source data are provided with this paper.

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

## Acknowledgements

We thank Jeff Dangl and H. Yoshioka for critical reading of this manuscript, M. Endo for the CRISPR/Cas9 expression constructs, Y. Nishizawa for *oscerk1-ko* mutant, K. Taoka for the split NanoLuc luciferase assay constructs, S. Kitano for technical assistance, T. Fukamiza and T. Ohnuma for chitin, T. Nakazaki and K. Nishimura for use of their greenhouse, and members of the Kawasaki Laboratory for technical assistance and discussions. This research was supported by Grants-in-Aid for Scientific Research (A) (19H00945), for Scientific Research on Innovative Areas (18H04789), for Exploratory Research (20K21320), Strategic International Collaborative Research project promoted by the Ministry of Agriculture, Forestry and Fisheries, Tokyo, Japan (JPJ0088379) and Basic Science Research Projects from the Mitsubishi Foundation to T.K.; by Grants-in-Aid for Scientific Research (JP15K18649) and Basic Science Research Projects from the Sumitomo Foundation to K. Yamaguchi, and by Grants-in-Aid for Scientific Research (B)(20H03191) and for Scientific Research on Innovative Areas (19H04856) to C.K. Diffraction data were collected at the Osaka University beamline BL44XU at SPring-8 (Harima, Japan) (Proposal No. 2015A6500).

## Author contributions

K.Y., K.H., C.K., and T.K. designed the project and experiments. K.Ichimaru, K.Y., Y.N., M.H., K.Ishikawa., H.I., S.S., K.Inoue, K.S., and S.Y. performed genetic, biochemical, and cell biology experiments and analyzed bacterial blight resistance. T.T. analyzed blast resistance. K.H., E.Y., T.F., A.N., and C.K. determined the crystal structure of PBI1. K.Y., K.H., C.K., and T.K. wrote the paper with input from all authors.

## Competing interests

The authors declare no competing interests.
