## [Peer Review File · Nature Communications]

Cooperative regulation of PBI1 and MAPKs controls WRKY45 transcription factor in rice immunityREVIEWER COMMENTS

Reviewer #1 (Remarks to the Author):

Protein homeostasis, which is often regulated by E3 ubiquitin ligases, plays a crucial role in regulating signaling intensity in diverse physiological responses, including plant immunity. This lab has previously shown that OsPUB44, a plant U-box containing E3 ubiquitin ligase, positively regulates microbial pattern-triggered immunity (PTI) in rice and is targeted by the virulence effector XopP secreted from the bacterial pathogen *Xanthomonas oryzae* (Ishikawa et al. Nature communications. 2014). In this manuscript, the authors presented data revealing the mechanism underlying OsPUB44 in PTI and being targeted by the virulence effector proteins. From a yeast-two hybrid screen, the authors have identified OsPBI1, OsPUB44-Interacting protein 1, which is a previously uncharacterized DUF1110 domain-containing protein. The crystal structure analysis of PBI1 revealed that OsPBI1 consists of a four-helix bundle structure, which bears a degree of similarities to the cc domains of plant cc-NB-LRRs, including Rx and MLA10. This is potentially interesting but was not further developed in this manuscript. The nuclear localization of OsPBI1 prompted the authors to test OsPBI1 interaction with transcriptional regulators. They found that OsPBI1 interacts with OsWRKY45 in the nucleus, which has been demonstrated to be required for rice immunity against rice blast and bacterial blight diseases. Here, they showed that OsPBI1 negatively regulates the protein level of OsWRKY45 and OsWRKY45 plays a role in chitin-mediated immune responses. In addition, the OsPBI1 homeostasis is mediated by OsPUB44 while the OsPBI1 protein level is reduced upon chitin perception, which is accompanied with the release of OsWRKY45, thereby improving the immunity response. Furthermore, OsMPK-mediated OsWRKY45 phosphorylation is important for the association between OsWRKY45 and OsPBI1. Overall, the data indicated layered regulations of OsWRKY45 protein complex homeostasis by phosphorylation, ubiquitination, and complex association and dissociation in rice immunity. The authors have presented some interesting observations which provide new insight into how OsPUB44 regulates OsWRKY45 by interacting with OsPBI1 in chitin-mediated immunity and disease resistance in rice. The genetic data showing that OsPBI1 negatively regulates rice disease resistance are solid. However, some of the conclusions derived from biochemical data in particular on OsPUB44 interactions with OsPBI1 and the connections of OsWRKY45 with OsPBI1 need additional justifications. Some of the conclusions lack sufficient data to support (see below).

1) "PUB44 interacts with PBI1" (line 12, page 7). The authors have performed yeast two-hybrid assays to confirm PUB44 interaction with PBI1 (Fig 1b-c). This needs alternative methods, including in vivo co-IP or in vitro pull-down assays, to support the claim. In addition, Co-IP assays will address whether the association of PUB44 and PBI1 is affected upon chitin treatment.

2) Does bacterial PAMP PGN induce the degradation of PBI1 since the *pbi1* mutants are resistant to the bacterial pathogen *Xanthomonas oryzae* pv. *oryzae* (Fig. 5e-g)? Are the *pbi1* mutants resistant to the fungal pathogens, such as *Magnaporthe oryzae*, since the authors mainly studied fungal chitin-mediated responses in rice in the manuscript?

3) "PBI1 degradation may be regulated by the PUB44-mediated ubiquitination pathway" (line 1, page 10). The authors have no data on the ubiquitination of PBI1. The authors need to test the ubiquitination of PBI1 upon chitin perception in WT and *pub44* mutant rice in order to make such a claim.

4) "PBI1 interacts with WRKY45 in the nucleus" (line 7, page 11). The authors have performed BiFC and Co-IP assays to support that PBI1 interacts with WRKY45. However, these assays could not rule out the possibility of indirect interactions. To claim this, the authors need perform either in vitro pull-down or yeast two-hybrid assays. The association of PBI1 and WRKY45 also should be tested with chitin treatment since the association of PBI1 to WRKY45 is crucial for gene transcription upon chitin perception.

5) The authors elucidated the structure of PBI1 which shows a high degree of similarity with the Rx CC domain of CC-NB-LRR. PBI1 contains four helices. Are they important for the function of PBI1, including its association with PUB44 and WRKY45?

6) Myc-WRKY45a showed two bands in the first and third line WB of Fig. 4b. The upper band should be phosphorylation bands (Ueno et al. PLoS Pathogens. 2015). Could PBI1 associate with the phosphorylated WRKY45a, which is not consistent with Fig. 6d that "Phosphorylation of WRKY45

inhibits the interaction between PBI1 and WRKY45"? Does PBI1 inhibit the phosphorylation of WRKY45, which is crucial for its transcription activity since the phosphorylation band of WRKY45 is stronger in *pbi1* mutants (Fig 5c)? Do *mapkkk11/mapkkk18* mutants reduce the phosphorylation of WRKY45? Does PBI1 associate with WRKY45 in *mapkkk11/mapkkk18* mutants? Additional evidence is needed to support that "Phosphorylation of WRKY45 inhibits the interaction between PBI1 and WRKY45" (Fig. 6d).

Specific comments:

1. The authors have performed a yeast two-hybrid screen for proteins that interact with the ARM domain of PUB44. The authors need to provide the whole list of candidates from the screen and those that were further confirmed in addition to PBI1.
2. Did the authors test (GluNAc)⁷⁻ and PGN-induced immune response, including MAPK activation and the defense gene expression, and the pathogen resistance in *pub44* mutants?
3. PBI1 and WRKY45 predominantly locate in the nucleus (fig. 4a) while PUB44 locates in the cytoplasm (Ishikawa et al. Nature communications. 2014). How does the PUB44-mediated PBI1 regulate the activity of WRKY45 in the nucleus?
4. *pbi1* mutants grow smaller than WT rice in figure 5b. Do these mutants show auto-immunity and cell death? Do these mutants have elevated PR1/2 expression and SA level?
5. Did the authors test the association between WRKY45 and PUB44 since WRKY45 is regulated by ubiquitination in rice (Matsushita et al. Plant Journal. 2013)? Does the ubiquitination of WRKY45 change in *pbi1* mutants since the WRKY45 protein level is accumulated in *pbi1* mutants (Fig. 5c)?
7. It is not clear that the λ -phosphatase dephosphorylates PUB44 in Fig. 7b. It will be better to contain the time point at 30 min which the phosphorylation of PUB44 was attenuated as control (Fig. 7c).
8. In the discussion, the authors have no enough evidence to support the conclusion "The phosphorylation of WRKY45....(line 6, page 19)" "This stimulates the release of WRKY45 from PBI1. At the same time, PUB44 is phosphorylated and then PBI1 is degraded, possibly following the disassociation from WRKY45. (line 16, page 19)" The authors should tune down these claims.

Reviewer #2 (Remarks to the Author):

The study "Cooperative regulation of PBI1 and MAPKs precisely controls the master transcription factor WRKY45 in rice immunity" by Ichimaru et al. identifies the protein PBI1, which is proposed to play a role in the regulation of WRKY45 and to be targeted by PUB44 to regulate immune responses. Authors provide data supporting a potential interaction between PBI1 and the E3 ligase PUB44. They investigate the role of PUB44 in the degradation of PBI1, as well as of the bacterial effector XopP, which was previously shown to inhibit PUB44 activity. Subsequently, authors investigate the interaction of PBI1, which they show displays a nucleo-cytoplasmic localization, with the transcription factor WRKY45. They show that in addition to interacting with WRKY45, PBI1 inhibits WRKY45-mediated transactivation. They show that *pbi1* mutants are more susceptible to a bacterial pathogen. They provide data, which supports a role of MAPK cascade activation in the inhibition of PBI1-WRKY45 interaction. Finally, they show that PUB44 is likely phosphorylated after immunostimulation. Based on these results, authors propose a model in which PBI1 negatively regulates WRKY45 function. Negative regulation is suggested to be relieved by phosphorylation of WRKY45, which potentially makes it accessible to PUB44 for degradation. In all, the manuscript provides several very interesting insights. However, several of the data still need further confirmation and some do not support the authors claims. Parts of the story seem fragmented, as they lack a clear link to the overall working model. I have various suggestions for the authors to consider.

Major points

1. The relationship between PUB44 and PBI1 is unclear. The interaction between PUB44 and PBI1 needs to be better characterized, *in vivo* data is required to demonstrate a true interaction which is also physiologically relevant. Ideally authors should also demonstrate a physical interaction between component PUB44-PBI1-WRKY45 to better understand the relationship between them, by *in vitro* assays.

Authors propose that PUB44 mediates the degradation of PBI1, however, the provided data in Figure 2 does not support this assumption. PUB44-kd displays reduced amounts of PBI1, which is opposite to the expected, namely an accumulation of PBI1 due to the lack of PUB44 ubiquitination and degradation. Of note, pub44 mutants display a reduced expression of PBI1, suggesting that the observed effect is rather due to transcriptional inhibition and not protein degradation. This would also hold true for Figure 2d, as a transcriptional inhibition after chitin treatment would be lost in the PUB44-kd. Moreover, the slow effect of MG132 rather suggests a slow turnover of the protein. These results therefore, do not support the authors' favoured hypothesis. To solve this inconsistencies it will be necessary to demonstrate that first PUB44 interacts with PBI1, and second that it mediates its ubiquitination. Ideally, authors should perform an in vitro ubiquitination assay, show that the ubiquitination levels of PBI1 are dependent on PUB44, and that PBI1 degradation rate is reduced in PUB44-kd.

Along the same lines, authors' data supporting MAPK-dependent degradation of PBI1 is not convincing. Figure 6b and 6e again supports a role of MAPK signalling in the transcriptional regulation of PBI1. The blots indicate reduced protein levels at time 0 for both alleles and in allele #2, rather what seems a reduction of the protein levels.

2. The potential role of PBI1 in regulating WRKY45 is very interesting, but requires further investigation. Authors should test whether PBI1 is also able to interact with other WRKY TFs, and whether there is specificity to WRKY45. Also, please include appropriate controls such as homologous proteins or mutants that do not interact, as well as blots showing protein expression.

In addition, further confirmation of PBI1's role is required. One key question to learn more about its mode of action would be to determine whether PBI1 inhibits WRKY45 binding to WW boxes.

3. The claim that WRKY45 participates in PTI is not supported by the data. Authors only provide data regarding the transcriptional induction after chitin treatment. To prove that WRKY45 really participates in PTI, needs to be demonstrated experimentally e.g. by using mutants (or KDs) to show an effect on PTI responses/resistance.

4. Authors should try to connect the observation that PUB44 is likely phosphorylated by MAPKs to the rest of the story by showing that PBI1 ubiquitination levels are affected in mapkkk11-1/mapkkk18. Figure 7b also needs improvement; the reduced signal of the shifted band correlates to the reduced signal of the non-shifted band, which declines gradually, suggesting a substrate problem.

Other points

-PBI1 characterization of the Ab is insufficient as shown in Figure1 is not sufficient. Authors should include the data provided in Figure 5a.

-Include size marker for all blots.

-What does the homology between PBI1 structure with that of CC from NLR tell us? Authors need to better integrate their observations into the current knowledge of NLR biology.

-In Figure 2b authors claim no transcriptional induction of PBI1, but it does actually look significant at 6h. What test was employed to determine significance?

-Figure 3C the localization is not predominantly in the nucleus. A significant portion is in the cytoplasm.

-Authors state that WRKY45 was identified from a screen. Please elaborate on what type of screen and how was it performed. Authors also need to disclose information regarding the screen that led to the discovery of PBI1, which is not even included in the M&M.

- In Figure 4B, GFP runs at 48kDa although its size is 27kDa. Please comment.

-It is unclear why enhanced activation of WRKY62 in pbi1 suggests the existence of an additional component. If WRKY45 is upstream and hyperactive in pbi1, wouldn't it be expected to also induce WRKY62.

Reviewer #3 (Remarks to the Author):

In this manuscript, Ichimaru et al perform novel functional analysis of the rice WRKY transcription factor WRKY45. They identify two interlinked systems, (i) identification and characterization of a novel interactor of PUB44, PBI1; PBI1 inhibits the activity of WRKY45 prior to immune elicitation, PBI1 is degraded by PUB44 on perception of chitin, (ii) characterization of MAPK-dependent phosphorylation of both PUB44 (promoting its activity) and WRKY45 directly (reducing affinity for PBI1) on perception of chitin. Cooperation between these pathways enables WRKY45 to activate immunity-related transcription events in rice.

Overall this is well-conducted research, and a well-written manuscript. Although at times the different sections seem a little detached from each other, the authors do bring together the different directions of the project into a coherent narrative. I consider that the paper will be of interest to others investigating the molecular basis of plant immunity and plant:microbe interactions.

Major considerations:

1. Please move Extended Data Fig. 4 to the main manuscript. This is a somewhat complicated manuscript with interconnected results that challenges the reader to string together a number of experiments, approaches, and proteins. An overarching model presented in the main manuscript will significantly enhance understanding.
2. Did the authors use PBI3 or PBI4 in any of their experiments as a negative control? Having demonstrated the lack of interaction of these proteins with PUB44, this would have been better than using empty vector, free GFP or GUS etc as controls in various experiments (perhaps in particular the co-IP and BiFC of Fig. 4). To be clear, I don't expect experiments to be re-done, but if this data exists it should be included.
3. p11 - the authors state "We screened for rice factors that interact with PBI1 and identified WRKY45 as a candidate". I cannot see any further details of this screen in the manuscript, or a reference to previous work. Additional info needs to be provided to show how the PBI1-WRKY45 interaction was initially identified.
4. While technically sound, the structure of PBI1 does not really contribute to the manuscript in a substantive way to understanding biology. Perhaps the authors could elaborate further on any structural features that hint at function, for example a region that maybe important for PUB44 or WRKY45 interaction?
5. Where appropriate, please include a statement about experimental repetitions. e.g. how many times were Y2H, co-IP, BiFC, rice infections, etc done. Were consistent results obtained?
6. I am concerned about the blots/ponceaus of Fig 7a. The gaps between the time courses and the last lane do not seem of a consistent size with the other lanes. Also, for the lower panel, the pub44-1 sample and ponceau lane sizes seem to not match. Could the authors provide the uncropped full blots for reassurance?
7. Where appropriate, I encourage the authors to present their data in the form of box-plots rather than bar graphs.

Minor comments:

1. Use NLR rather than NB-LRR as an abbreviation throughout.
2. All blots presented in the figures throughout the manuscript should include size markers (including ponceau staining, or indicate the likely protein stained (e.g. Rubisco)).

3. It would be useful for the authors to provide a protein sequence alignment of the PBI proteins, and annotate structural elements on this (can be in the SI). Does such an alignment show a region of the proteins that might be important for PUB44 or WRKY45 binding?

4. For the initial Y2H analysis. Why is 3AT used in some experiments, but not others?

We thank the reviewers for careful review as well as constructive comments. According to the reviewers' suggestions, we have thoroughly revised the manuscript. Below, we provide a detailed point-by-point response addressing the reviewers' criticisms.

Reviewer #1:

Protein homeostasis, which is often regulated by E3 ubiquitin ligases, plays a crucial role in regulating signaling intensity in diverse physiological responses, including plant immunity. This lab has previously shown that OsPUB44, a plant U-box containing E3 ubiquitin ligase, positively regulates microbial pattern-triggered immunity (PTI) in rice and is targeted by the virulence effector XopP secreted from the bacterial pathogen Xanthomonas oryzae (Ishikawa et al. Nature communications. 2014). In this manuscript, the authors presented data revealing the mechanism underlying OsPUB44 in PTI and being targeted by the virulence effector proteins. From a yeast-two hybrid screen, the authors have identified OsPBII, OsPUB44-Interacting protein 1, which is a previously uncharacterized DUF1110 domain-containing protein. The crystal structure analysis of PBII revealed that OsPBII consists of a four-helix bundle structure, which bears a degree of similarities to the cc domains of plant cc-NB-LRRs, including Rx and MLA10. This is potentially interesting but was not further developed in this manuscript. The nuclear localization of OsPBII prompted the authors to test OsPBII interaction with transcriptional regulators. They found that OsPBII interacts with OsWRKY45 in the nucleus, which has been demonstrated to be required for rice immunity against rice blast and bacterial blight diseases. Here, they showed that OsPBII negatively regulates the protein level of OsWRKY45 and OsWRKY45 plays a role in chitin-mediated immune responses. In addition, the OsPBII homeostasis is mediated by OsPUB44 while the OsPBII protein level is reduced upon chitin perception, which is accompanied with the release of OsWRKY45, thereby improving the immunity response. Furthermore, OsMPK-mediated OsWRKY45 phosphorylation is important for the association between OsWRKY45 and OsPBII. Overall, the data indicated layered regulations of OsWRKY45 protein complex homeostasis by phosphorylation, ubiquitination, and complex association and dissociation in rice immunity. The authors have presented some interesting observations which provide new insight into how OsPUB44 regulates OsWRKY45 by interacting with OsPBII in chitin-mediated immunity and disease resistance in rice. The genetic data showing that OsPBII negatively regulates rice disease resistance are solid. However, some of the conclusions derived from biochemical data in particular on OsPUB44 interactions with OsPBII and the connections of OsWRKY45 with OsPBII

need additional justifications. Some of the conclusions lack sufficient data to support (see below).

Reply: We thank the reviewer for careful review as well as positive and constructive comments.

Reviewer 1 comment

1-1) "PUB44 interacts with PBI1" (line 12, page 7). The authors have performed yeast two-hybrid assays to confirm PUB44 interaction with PBI1 (Fig 1b-c). This needs alternative methods, including in vivo co-IP or in vitro pull-down assays, to support the claim. In addition, Co-IP assays will address whether the association of PUB44 and PBI1 is affected upon chitin treatment.

Answer: We thank the reviewer for constructive comments. According to the comment, we performed Co-IP assay (Figure 1e), *in vitro* pull down assay (Figure 1g), and Split NanoLuc Luciferase assay (Figure 1f) to analyze the interaction between PUB44 and PBI1. These results indicated that PBI1 directly interacts with PUB44 *in vitro* and *in vivo*. We added these results in the manuscript. As pointed out by the reviewer, whether the interaction between PBI1 and PUB44 is altered during chitin response is an important question. However, immunoprecipitation with α -PUB44 and α -PBI1 did not work because of low reactivity of these antibodies. Therefore, to analysis the *in vivo* interaction between PUB44 and PBI1 during chitin response, we need an experimental system using the protoplasts transiently expressing epitope-tagged PUB44 and PBI1. However, in this system, rice protoplasts do not respond to chitin, although same experiment can work in Arabidopsis system. To our knowledge, no paper showed the evidence that chitin responses occur in rice protoplast. One possibility is that cell wall-derived debris produced during preparation of protoplast may inhibit the chitin responses of the protoplasts. For these reasons, we could not examine whether the interaction between PBI1 and PUB44 is altered during chitin response.

Reviewer 1 comment

*1-2) Does bacterial PAMP PGN induce the degradation of PBI1 since the *pbi1* mutants are resistant to the bacterial pathogen *Xanthomonas oryzae* pv. *oryzae* (Fig. 5e-g)? Are the *pbi1* mutants resistant to the fungal pathogens, such as *Magnaporthe oryzae*, since the authors mainly studied fungal chitin-mediated responses in rice in the manuscript?*

Answer: We appreciate your valuable suggestion. According to the comment, we tested whether treatment with PGN induces PBI1 degradation. This result indicated that PBI1 was degraded upon perception of PGN, which is consistent with the fact that PUB44 functions downstream of OsCERK1 involving in PGN perception. We added the result (Figure 2c). We also analyzed blast resistance of the *pbi1* mutants using the compatible race *Magnaporthe oryzae* ken53-33 by collaboration with Dr. Takeda, an expert of *M. oryzae* disease. However, we did not detect significant difference between WT and *pbi1* mutants. We added the result (Supplementary Figure 10c). At the present, we do not know the reason why the *pbi1* mutants enhance blast resistance.

Reviewer 1 comment

1-3) “PBI1 degradation may be regulated by the PUB44-mediated ubiquitination pathway” (line 1, page 10). The authors have no data on the ubiquitination of PBI. The authors need to test the ubiquitination of PBI1 upon chitin perception in WT and *pub44* mutant rice in order to make such a claim.

Answer: We thank for your valuable suggestion. According to the comment, we examined ubiquitination of PBI1 by co-IP with α -Ubq and the immunoblot with α -PBI1. We found that high molecular weight bands that possibly correspond to poly-ubiquitinated PBI1 protein were increased by chitin treatment. Instead, the level of unmodified PBI1 protein was reduced as consistent with Figure 2a. The increase of the high molecular weight bands was not observed in the PUB44 knockdown cell, suggesting that PBI1 ubiquitination occurs dependent upon PUB44. We added the result in the manuscript (Figure 2f).

Reviewer 1 comment

1-4) “PBI1 interacts with WRKY45 in the nucleus” (line 7, page 11). The authors have performed BiFC and Co-IP assays to support that PBI1 interacts with WRKY45. However, these assays could not rule out the possibility of indirect interactions. To claim this, the authors need perform either *in vitro* pull-down or yeast two-hybrid assays. The association of PBI1 and WRKY45 also should be tested with chitin treatment since the association of PBI1 to WRKY45 is crucial for gene transcription upon chitin perception.

Answer: We thank for your valuable advice. According to the comment, we examined direct interaction between PBI1 and WRKY45 by *in vitro* pull down assay. The result indicated that PBI1 directly interacts with WRKY45. We added the result (Figure 4c). Whether chitin treatment affects the association between PBI1 and

WRKY45 is an important question. However, the level of endogenous WRKY45 protein is very low, and immunoprecipitation with α -WRKY45 and α -PBI1 did not work because of low reactivity of these antibodies. Therefore, we need to use the protoplasts transiently expressing WRKY45 and PBI1 as shown in Figure 4b. However, as mentioned above, in available experimental methods, rice protoplasts do not respond to chitin.

Reviewer 1 comment

1-5) The authors elucidated the structure of PBI1 which shows a high degree of similarity with the Rx CC domain of CC-NB-LRR. PBI1 contains four helices. Are they important for the function of PBI1, including its association with PUB44 and WRKY45?

Answer: We thank for your question. WRKY45 is known to interact with the CC domain of Pb1, rice CC-NB-LRR protein (Hayashi et al. Plant J). Although the structure of the Pb1 CC domain was not determined, Pb1 might contain four helices based on the information of the Rx CC domain. However, we do not know whether the helices is important to associate with PUB44 and WRKY45.

Reviewer 1 comment

1-6) Myc-WRKY45a showed two bands in the first and third line WB of Fig. 4b. The upper band should be phosphorylation bands (Ueno et al. PLoS Pathogens. 2015). Could PBI1 associate with the phosphorylated WRKY45a, which is not consistent with Fig. 6d that “Phosphorylation of WRKY45 inhibits the interaction between PBI1 and WRKY45”? Does PBI1 inhibit the phosphorylation of WRKY45, which is crucial for its transcription activity since the phosphorylation band of WRKY45 is stronger in pbi1 mutants (Fig 5c)? Do mapkkk11/mapkkk18 mutants reduce the phosphorylation of WRKY45? Does PBI1 associate with WRKY45 in mapkkk11/mapkkk18 mutants? Additional evidence is needed to support that “Phosphorylation of WRKY45 inhibits the interaction between PBI1 and WRKY45” (Fig. 6d).

Answer: We appreciate your valuable questions. We always detected two bands of WRKY45 without any stimulation, even when the WRKY45 protein was produced using wheat germ *in vitro* protein expression system (Figure 4c). In addition, the levels of two bands were consistent, which was also observed in other reports (Matsushita et al. Plant J 2013). Therefore, we do not know biological importance of the two bands of WRKY45 detected in unstimulated conditions. Although MAPKs have been reported to phosphorylate WRKY45 by *in vitro* experiments, the WRKY45 proteins phosphorylated by MAPKs *in vivo* have not been observed.

Therefore, it is possible that the band corresponding to the WRKY45 protein phosphorylated by MAPKs may be different from the two bands detected in unstimulated condition. Because the WRKY45 protein levels in leaves were hardly detectable in wild type as shown in Figure 5c, comparison of the phosphorylated levels of WRKY45 is technically difficult. In addition, the immunoblots with α -WRKY45 showed high background when we used proteins purified from suspension cells. Thus, it is difficult for us to analyze MAPK-mediated phosphorylation of WRKY45. Therefore, we toned down the statement concerning that phosphorylation of WRKY45 inhibits the interaction between PBI1 and WRKY45 (Abstract, p18 line 19-21, p22 line 16 – p23 line 7).

Reviewer 1 comment

1-7. The authors have performed a yeast two-hybrid screen for proteins that interact with the ARM domain of PUB44. The authors need to provide the whole list of candidates from the screen and those that were further confirmed in addition to PBI1.

Answer: We thank for your valuable suggestions. We isolated two other positive clones by initial screening. We added the information (Supplemental Figure 1). However, the interaction of these candidates with PUB44 was not supported by other methods including BiFC (p8 line 8-11).

Reviewer 1 comment

1-8. Did the authors test (GluNAc)₇- and PGN-induced immune response, including MAPK activation and the defense gene expression, and the pathogen resistance in *pub44* mutants?

Answer: We thank for your comment. In the previous paper, we showed that *PUB44*-knockdown (*PUB44-kd*) did not affect chitin-induced MAPK activation (Supplemental data of Ishikawa et al. Nat Commun 2014). The defense gene expression and the pathogen resistance were also described in the paper, indicating that *PUB44-kd* reduced the defense gene expression and the resistance to *X. oryzae*. In initial submission, the manuscript contained a few results of the *PUB44* knockout (*PUB44-ko*) mutants. However, during this revision, we realized that *PUB44-ko* exhibits different phenotype from *PUB44-kd*. For example, although *PUB44-kd* did not affect *PBI1* expression, *PUB44-ko* reduced the transcript levels of *PBI1*. These results suggest that complete loss of *PUB44* protein might induce additional responses as observed in other proteins such as BAK1 (Yamada et al (2016) EMBO J. 35, 46-61). Therefore, we need a series of experiments to understand biological

nature of *PUB44-ko*. Since this is a different topic, we deleted all data of the *PUB44-ko* mutants in the revised manuscript.

Reviewer 1 comment

1-9. *PBI1* and *WRKY45* predominantly locate in the nucleus (fig. 4a) while *PUB44* locates in the cytoplasm (Ishikawa et al. Nature communications. 2014). How does the *PUB44*-mediated *PBI1* regulate the activity of *WRKY45* in the nucleus?

Answer: We appreciate your valuable advice. We analyzed subcellular localization of *PUB44*-GFP by optical sectioning using a fluorescence microscope with Apotome2 system (Carl Zeiss). This result indicated that *PUB44*-GFP localized to both nucleus and cytoplasm. We added the data (Supplementary Figure 6).

Reviewer 1 comment

1-10. *pbi1* mutants grow smaller than WT rice in figure 5b. Do these mutants show auto-immunity and cell death? Do these mutants have elevated *PR1/2* expression and SA level?

Answer: We thank for your valuable comment. Although the *pbi1* mutants exhibit dwarf phenotype, this phenotype is weaker as compared with typical auto-immune phenotypes. In addition, we did not observe cell death. According to the comment, we analyzed expression of many defense genes. We only found upregulation of *PR10*, one of the *PR* genes, in the *pbi1* mutants (Supplementary Figure 10b). However, the *pbi1* mutation did not affect expression of other *PR* genes. Since *WRKY45* is known to regulate immune priming (Akagi et al. Plant Mol Biol. 2014, 86:171), it is possible that the *pbi1* mutation may induce immune priming through increase of the *WRKY45* protein levels. We added the explanation in the text (p16 line 16 – 21). Since rice contains high amount of SA without any stimuli, it is difficult to compare SA levels.

Reviewer 1 comment

1-11. Did the authors test the association between *WRKY45* and *PUB44* since *WRKY45* is regulated by ubiquitination in rice (Matsushita et al. Plant Journal. 2013)? Does the ubiquitination of *WRKY45* change in *pbi1* mutants since the *WRKY45* protein level is accumulated in *pbi1* mutants (Fig. 5c)?

Answer: We appreciate your valuable comment. According to your comment, we analyzed the interaction between *PUB44* and *WRKY45* by an *in vitro* pull down assay and a split NanoLuc luciferase assay. We detected the direct interaction

between PUB44 and WRKY45 by the *in vitro* pull down assay (Supplementary Figure 8a). However, the interaction was very weak, because the interaction was lost by washing with the buffer containing 0.1 % Triton-X (Supplementary Figure 8a). In addition, the split NanoLuc luciferase assay indicates that the interaction between PUB44 and WRKY45 was much weaker than the interaction between PUB44 and PBI1 (Supplemental Figure 8b). Therefore, these data suggest that PUB44 might not be involved in the WRKY45 ubiquitination (p14 line 12 – 20). In the previous report (Matsushita et al. 2013), the ubiquitinated WRKY45 proteins were detected by co-IP using transgenic cells overexpressing Myc-tagged WRKY45 under non-elicited condition, only when they used proteasome inhibitor MG132. As shown in Fig 5, since the protein level of WRKY45 was undetectable in wild type by immunoblot with α -WRKY45, it is difficult to compare ubiquitination level of WRKY45 between wild type and *pbi1*. Although it is possible that the ubiquitination of WRKY45 may be inhibited in the *pbi1* mutants, we thought that increase of the WRKY45 protein levels in the *pbi1* mutant is associated with transcription of WRKY45 because the transcript levels of *WRKY45* were increased in the *pbi1* mutants.

Reviewer 1 comment

1-12. It is not clear that the λ -phosphatase dephosphorylates PUB44 in Fig. 7b. It will be better to contain the time point at 30 min which the phosphorylation of PUB44 was attenuated as control (Fig. 7c).

Answer: We thank for your advice. According to your comment, we re-analyzed it and replaced it by new result (Figure 7b).

Reviewer 1 comment

1-13. In the discussion, the authors have no enough evidence to support the conclusion “The phosphorylation of WRKY45...(line 6, page 19)” “This stimulates the release of WRKY45 from PBI1. At the same time, PUB44 is phosphorylated and then PBI1 is degraded, possibly following the disassociation from WRKY45. (line 16, page 19)” The authors should tune down these claims.

Answer: We appreciate your valuable advice. We deleted or toned down the sentence (p22 line 16 – p23 line 7).

Reviewer #2 (Remarks to the Author):

*The study “Cooperative regulation of PBII and MAPKs precisely controls the master transcription factor WRKY45 in rice immunity” by Ichimaru et al. identifies the protein PBII, which is proposed to play a role in the regulation of WRKY45 and to be targeted by PUB44 to regulate immune responses. Authors provide data supporting a potential interaction between PBII and the E3 ligase PU44. They investigate the role of PUB44 in the degradation of PBII, as well as of the bacterial effector XopP, which was previously shown to inhibit PUB44 activity. Subsequently, authors investigate the interaction of PBII, which they show displays a nucleo-cytoplasmic localization, with the transcription factor WRKY45. They show that in addition to interacting with WRKY45, PBII inhibits WRKY45-mediated transactivation. They show that *pbi1* mutants are more susceptible to a bacterial pathogen. They provide data, which supports a role of MAPK cascade activation in the inhibition of PBII-WRKY45 interaction. Finally, they show that PUB44 is likely phosphorylated after immunostimulation. Based on these results, authors propose a model in which PBII negatively regulates WRKY45 function. Negative regulation is suggested to be relieved by phosphorylation of WRKY45, which potentially makes it accessible to PUB44 for degradation. In all, the manuscript provides several very interesting insights. However, several of the data still need further confirmation and some do not support the authors claims. Parts of the story seem fragmented, as they lack a clear link to the overall working model. I have various suggestions for the authors to consider.*

Reply: We thank the reviewer for careful review as well as positive and constructive comments.

Reviewer 2 comment

2-1. The relationship between PUB44 and PBII is unclear. The interaction between PUB44 and PBII needs to be better characterized, in vivo data is required to demonstrate a true interaction which is also physiologically relevant. Ideally authors should also demonstrate a physical interaction between component PUB44-PBII-WRKY45 to better understand the relationship between them, by in vitro assays.

Answer: We thank the reviewer for constructive comments. According to the comment, we performed Co-IP assay (Figure 1e), in vitro pull down assay (Figure 1g), and Split NanoLuc Luciferase assay (Figure 1f) to analyze the interaction between PUB44 and PBII. These results indicated that PBII directly interacts with PUB44 *in vivo* and *in vitro*. We added these results in the manuscript. We also

detected direct interaction between PBI1 and WRKY45 by *in vitro* pull down assay. We added the result (Figure 4c). In addition, we analyzed the interaction between PUB44 and WRKY45 by an *in vitro* pull down assay and a split NanoLuc luciferase assay. We detected the direct interaction between PUB44 and WRKY45 by the *in vitro* pull down assay (Supplementary Figure 8a). However, the interaction was very weak, because the interaction was lost by washing with the buffer containing 0.1 % Triton-X (Supplementary Figure 8a). In addition, the split NanoLuc luciferase assay indicates that the interaction between PUB44 and WRKY45 was much weaker than the interaction between PUB44 and PBI1 (Supplementary Figure 8b). Therefore, we could not find biological significance of the interaction between PUB44 and WRKY45.

Reviewer 2 comment

2-2) Authors propose that PUB44 mediates the degradation of PBI1, however, the provided data in Figure 2 does not support this assumption. PUB44-kd displays reduced amounts of PBI1, which is opposite to the expected, namely an accumulation of PBI1 due to the lack of PUB44 ubiquitination and degradation. Of note, pub44 mutants display a reduced expression of PBI1, suggesting that the observed affect is rather due to transcriptional inhibition and not protein degradation. This would also hold true for Figure 2d, as a transcriptional inhibition after chitin treatment would be lost in the PUB44-kd.

Answer: We appreciate your valuable advice. We re-analyzed expression of PBI1 in wild type and PUB44-kd cells. The PBI1 expression levels was slightly higher in PUB44-kd as compared with wild type (Supplementary Figure 4a). In addition, the expression levels of PBI1 in wild type and PUB44-kd cells were not changed by chitin treatment (Figure 2b and Supplementary Figure 4b). Therefore, it is likely that chitin-induced reduction of PBI1 protein levels was caused by protein degradation, and loss of PBI1 degradation in the PUB44-kd cell does not result from loss of transcriptional inhibition of PBI1. In initial submission, the manuscript contained a few results of the PUB44 knockout (PUB44-ko) mutants. However, during this revision, we realized that PUB44-ko exhibits different phenotype from PUB44-kd. For example, although PUB44-kd did not affect PBI1 expression, PUB44-ko reduces the transcript levels of PBI1. These results suggest that complete loss of PUB44 proteins might induce additional responses as observed in other proteins such as BAK1 (Yamada et al (2016) EMBO J. 35, 46-61). Therefore, we need a series of experiments to understand biological nature of PUB44-ko. Since this is a different topic, we deleted all data of the PUB44-ko mutants in the revised manuscript.

Reviewer 2 comment

2-3) Moreover, the slow effect of MG132 rather suggests a slow turnover of the protein. These results therefore, do not support the authors' favoured hypothesis. To solve this inconsistencies it will be necessary to demonstrate that first PUB44 interacts with PBI1, and second that it mediates its ubiquitination. Ideally, authors should perform an *in vitro* ubiquitination assay, show that the ubiquitination levels of PBI1 are dependent on PUB44, and that PBI1 degradation rate is reduced in PUB44-kd.

Answer: We appreciate your valuable comments. According to the comment, we examined ubiquitination of PBI1 by co-IP with α -Ubq and the immunoblot with α -PBI1. We found that high molecular weight bands that possibly correspond to poly-ubiquitinated PBI1 protein were increased by chitin treatment. Instead, the level of unmodified PBI1 protein was reduced as consistent with Figure 2a. The increase of the high molecular weight bands was not observed in the *PUB44-kd* cell, suggesting that PBI1 ubiquitination occurs dependent upon PUB44. We added the result in the manuscript (Figure 2f). In addition, we indicated the relative PBI1 protein levels in Figure 2a,c, e.g. These data indicate that PBI1 degradation rate was reduced in *PUB44-kd* cell compared with wild type. As described in our previous report (Ishikawa et al. Nat Commun 2014), recombinant full length PUB44 protein possesses only a faint ubiquitin ligase activity as compared with the U-box domain. Therefore, we could not detect ubiquitination of PBI1 by the *in vitro* ubiquitin ligase assay using full length PUB44 protein.

Reviewer 2 comment

2-4) Along the same lines, authors' data supporting MAPK-dependent degradation of PBI1 is not convincing. Figure 6b and 6e again supports a role of MAPK signalling in the transcriptional regulation of PBI1. The blots indicate reduced protein levels at time 0 for both alleles and in allele #2, rather what seems a reduction of the protein levels.

Answer: We thank for your comment. We carried out quantitative real-time PCR experiment to analyze the transcript levels of *PBI1* in the *mapkkk11/mapkkk18* mutants. This result indicates that the *mapkkk11/mapkkk18* mutation did not affect expression of *PBI1*. We added the results in Supplementary Figure 11b. We performed the immune blot experiments using the *mapkkk11/mapkkk18* mutants more than 5 times. Based upon these results, we concluded that the PBI1 protein levels were not changed in the mutants.

Reviewer 2 comment

2-5). *The potential role of PBI1 in regulating WRKY45 is very interesting, but requires further investigation. Authors should test whether PBI1 is also able to interact with other WRKY TFs, and whether there is specificity to WRKY45. Also, please include appropriate controls such as homologous proteins or mutants that do not interact, as well as blots showing protein expression.*

Answer: We appreciate your valuable suggestion. According to the suggestion, we examined interaction between PBI1 and other three WRKYs using *in vitro* pull down assay. However, PBI1 did not interact with other WRKYs than WRKY45. We added the results (Fig 4c and Supplementary Figure 7a). In addition, we examined the interactions using split NanoLuc luciferase assay. The data also indicated that PBI1 interacted specifically with WRKY45 (Supplementary Figure 7b). Although we did not test all WRKYs, it is likely that PBI1 prefers to interact with WRKY45.

Reviewer 2 comment

2-6) *In addition, further confirmation of PBI1's role is required. One key question to learn more about its mode of action would be to determine whether PBI1 inhibits WRKY45 binding to WW boxes.*

Answer: We thank for your valuable advice. According to the comment, we carried out an electrophoresis mobility shift assay. The data indicated that PBI1 does not inhibit the DNA binding activity of WRKY45. We added the data in Figure 4g and Supplementary Figure 9c.

Reviewer 2 comment

2-7). *The claim that WRKY45 participates in PTI is not supported by the data. Authors only provide data regarding the transcriptional induction after chitin treatment. To proof that WRKY45 really participates in PTI, needs to be demonstrated experimentally e.g. by using mutants (or KDs) to show an effect on PTI responses/resistance.*

Answer: We thank for your advice. We indicate that expression of WRKY62, a known downstream gene of WRKY45, was down-regulated in WRKY45-kd cells (Figure 4e). We normally analyze MAPK activation and ROS production to test involvement in PTI. However, since the WRKY45 activation occurs downstream of MAPKs activation and ROS production, it is unlikely that WRKY45 is involved in these responses. Therefore, we removed the statement “WRKY45 participates in PTI”.

Reviewer 2 comment

2-8). Authors should try to connect the observation that PUB44 is likely phosphorylated by MAPKs to the rest of the story by showing that PBI1 ubiquitination levels are affected in *mapkkk11-1/mapkkk18*. Figure 7b also needs improvement; the reduced signal of the shifted band correlates to the reduced signal of the non-shifted band, which declines gradually, suggesting a substrate problem.

Answer: We thank for your comment. We are sorry to confuse you. Because the phosphorylation of PUB44 was detected at similar level at 10 min after chitin treatment in the *mapkkk11/mapkkk18* mutants, we do not think that MAPKs phosphorylate PUB44. In the discussion, we indicated “The phosphorylation of PUB44 was also observed in the *mapkkk11/mapkkk18* mutants, although it was delayed and reduced. Therefore, it is unlikely that MAPKs phosphorylate PUB44. The reduced level of phosphorylation may be explained by the fact that *OsCERK1* expression was reduced in the *mapkkk11/mapkkk18* mutants. The identification of protein kinases that phosphorylate PUB44 will be required for a further understanding of PUB44 activation”.

In addition, according to your comment, we re-analyzed the phosphorylation of PUB44, and replaced it by new result (Figure 7b).

Reviewer 2 comment

2-9)-PBI1 characterization of the Ab is insufficient as shown in Figure 1 is not sufficient. Authors should include the data provided in Figure 5a.

Answer: We thank for your comment. According to the comment, we added the data of the immunoblotting using proteins purified from suspension cell cultures of WT and the *pbi1* mutants (Supplementary Figure 3a,b). The data indicated that no proteins were detected in the *pbi1* cells by the immune blot with anti-PBI1 antibody.

Reviewer 2 comment

2-10)-Include size marker for all blots.

Answer: We thank for your advice. We added the size marker for all blots.

Reviewer 2 comment

2-11)-*What does the homology between PBI1 structure with that of CC from NLR tell us? Authors need to better integrate their observations into the current knowledge of NLR biology.*

Answer: We thank for your comment. PBI1 interacts with WRKY45, and WRKY45 is known to interact with the CC domain of Pb1, rice CC-NB-LRR protein (Hayashi et al. Plant J). Although the structure of the Pb1 CC domain was not determined, the structural similarity between PBI1 and the CC domain may explain the fact that WRKY45 interacts with both PBI1 and the CC domain of Pb1. However, we do not think that PBI1 possesses similar regulatory systems as CC-NLRs.

Reviewer 2 comment

2-12)-*In Figure 2b authors claim no transcriptional induction of PBI1, but it does actually look significant at 6h. What test was employed to determine significance?*

We thank for your comment. We re-analyze the expression of PBI1 during chitin response and replaced the data (Figure 2b). The Student's t-test analysis indicated no significant change of PBI1 expression during chitin treatment, although the levels were slightly increased. Since PBI1 degradation occurs before 60 min, we showed the data of 0 – 60 min.

Reviewer 2 comment

2-13)-*Figure 3C the localization is not predominantly in the nucleus. A significant portion is in the cytoplasm.*

Answer: We thank you for your comment. We replaced the statement to “Fluorescence from both the GFP-PBI1 and PBI1-GFP proteins was detected in nuclei and cytoplasm “.

Reviewer 2 comment

2-14)-*Authors state that WRKY45 was identified from a screen. Please elaborate on what type of screen and how was it performed. Authors also need to disclose information regarding the screen that led to the discovery of PBI1, which is not even included in the M&M.*

Answer: We thank for your advice. We added the information of the screening process for PBI1 and WRKY45 in the text (p13 line 6-8).

Reviewer 2 comment

2-15)- *In Figure 4B, GFP runs at 48kDa although its size is 27kDa. Please comment.*

Answer: We thank for your notice. We corrected the data.

Reviewer 2 comment

2-16)-*It is unclear why enhanced activation of WRKY62 in pbi1 suggests the existence of an additional component. If WRKY45 is upstream and hyperactive in pbi1, wouldn't it be expected to also induce WRKY62.*

Answer: We appreciate your comment. Because chitin-induced expression of WRKY62 was observed in the absence of PBI1, we wanted to suggest the existence of additional factor, that is the existence of MAPK-mediated regulation of WRKY45. However, because it was confusing, we deleted the sentence in the result and discussion sections.

Reviewer #3 (Remarks to the Author):

Reviewer 3 comment

In this manuscript, Ichimaru et al perform novel functional analysis of the rice WRKY transcription factor WRKY45. They identify two interlinked systems, (i) identification and characterization of a novel interactor of PUB44, PBII; PBII inhibits the activity of WRKY45 prior to immune elicitation, PBII is degraded by PUB44 on perception of chitin, (ii) characterization of MAPK-dependent phosphorylation of both PUB44 (promoting its activity) and WRKY45 directly (reducing affinity for PBII) on perception of chitin. Cooperation between these pathways enables WRKY45 to activate immunity-related transcription events in rice.

Overall this is well-conducted research, and a well-written manuscript. Although at times the different sections seem a little detached from each other, the authors do bring together the different directions of the project into a coherent narrative. I consider that the paper will be of interest to others investigating the molecular basis of plant immunity and plant:microbe interactions.

Reply: We thank the reviewer for careful review as well as positive and constructive comments.

Reviewer 3 comment

3-1. Please move Extended Data Fig. 4 to the main manuscript. This is a somewhat complicated manuscript with interconnected results that challenges the reader to string together a number of experiments, approaches, and proteins. An overarching model presented in the main manuscript will significantly enhance understanding.

Answer: We appreciate your valuable advice. We moved the proposed model to the main manuscript.

Reviewer 3 comment

3-2. Did the authors use PBI3 or PBI4 in any of their experiments as a negative control? Having demonstrated the lack of interaction of these proteins with PUB44, this would have been better than using empty vector, free GFP or GUS etc as controls in various experiments (perhaps in particular the co-IP and BiFC of Fig. 4). To be clear, I don't expect experiments to be re-done, but if this data exists it should be included.

Answer: We thank for your advice. However, we do not have such data.

Reviewer 3 comment

3-3. p11 - the authors state “We screened for rice factors that interact with PBI1 and identified WRKY45 as a candidate”. I cannot see any further details of this screen in the manuscript, or a reference to previous work. Additional info needs to be provided to show how the PBI1-WRKY45 interaction was initially identified.

Answer: We thank for your advice. We added the information of the screening process for PBI1 and WRKY45 in the text (p13 line 6-8).

Reviewer 3 comment

3-4. While technically sound, the structure of PBI1 does not really contribute to the manuscript in a substantive way to understanding biology. Perhaps the authors could elaborate further on any structural features that hint at function, for example a region that maybe important for PUB44 or WRKY45 interaction?

Answer: We thank for your comment. PBI1 interacts with WRKY45, and WRKY45 is known to interact with the CC domain of Pb1, rice CC-NB-LRR protein (Hayashi et al. Plant J). Although the structure of the Pb1 CC domain was not determined, the structural similarity between PBI1 and the CC domain may explain the fact that WRKY45 interacts with both PBI1 and the CC domain of Pb1. However, we did not obtain biological information from the structural features of PBI1.

Reviewer 3 comment

3-5. Where appropriate, please include a statement about experimental repetitions. e.g. how many times were Y2H, co-IP, BiFC, rice infections, etc done. Were consistent results obtained?

Answer: We thank for your valuable advice. We added the information of experimental repetitions in all figure legends.

Reviewer 3 comment

3-6. I am concerned about the blots/ponceaus of Fig 7a. The gaps between the time courses and the last lane do not seem of a consistent size with the other lanes. Also, for the lower panel, the pub44-1 sample and ponceau lane sizes seem to not match. Could the authors provide the uncropped full blots for reassurance?

Answer: We thank for your comment. We provide the uncropped full data for all blots (Supplementary Figure 12).

Reviewer 3 comment

3-7. Where appropriate, I encourage the authors to present their data in the form of box-plots rather than bar graphs.

Answer: We thank for your advice. We made Figure 5f using box-plots.

Reviewer 3 comment

3-8. Use NLR rather than NB-LRR as an abbreviation throughout.

Answer: We appreciate your advice. We replaced NB-LRR by NLR.

Reviewer 3 comment

3-9. All blots presented in the figures throughout the manuscript should include size markers (including ponceau staining, or indicate the likely protein stained (e.g. Rubisco)).

Answer: We appreciate your valuable suggestion. We added the size marker for all blots.

Reviewer 3 comment

3-10. It would be useful for the authors to provide a protein sequence alignment of the PBI proteins, and annotate structural elements on this (can be in the SI). Does such an alignment show a region of the proteins that might be important for PUB44 or WRKY45 binding?

Answer: We thank for your advice. We added the sequence alignment of PBI family in Supplementary Figure 2. However, we could not determine which region of PBI1 is responsible for the interaction with PUB44 or WRKY45.

Reviewer 3 comment

4. For the initial Y2H analysis. Why is 3AT used in some experiments, but not others?

Answer: We thank for your question. When the background was high, we used 3AT.

Reviewers' comments:

Reviewer #1 (Remarks to the Author):

The authors have performed additional experiments and edited writings. Most concerns were addressed. However, this reviewer still feels that the evidence linking PUB44-PBI and PBI1-WRKY45 in the chitin and PGN response is relatively weak. The authors explained that the antibodies against PUB44 and PBI1 did not work, which hampers their efforts in detecting the complex association in vivo. The authors have responded in the rebuttal letter that no paper showed the evidence that chitin responses occur in rice protoplasts. However, briefly checking research on the chitin perception system in rice, this reviewer has noticed reports that chitin could trigger the formation of OsLYP4/OsLYP6/OsCEBiP/OsCERK1 complexes and also the dissociation of OsCERK1 and OsRLCK176 in rice protoplast (Ao et al., 2014, The Plant Journal). Since PGN treatment works in rice protoplasts, the authors could also perform the complex formation assays using PGN. These experiments will explain and tighten up the relationship between OsPUB44, OsPBI1, and OsWRKY45 in chitin-triggered responses.

Figure 2F. The authors need to show the IP loading by α -Ubi western-blotting. Since they have the pbi1 mutant, they should include this as a negative control to confirm that the smear band is specific for ubiquitinated PBI1 in the Ubi assay.

Title: delete "Precisely" and "master". This reviewer feels no evidence to support such claims from this work.

Reviewer #3 (Remarks to the Author):

Having reviewed the resubmitted paper and the rebuttal I judge that the authors have responded to all of my comments satisfactorily.

We thank the editor and the reviewers for careful review as well as constructive comments. According to the editor and the reviewers' suggestions, we have thoroughly revised the manuscript. Below, we provide a detailed point-by-point response addressing the editor and reviewers' criticisms.

Reviewer #1

(1) The authors have performed additional experiments and edited writings. Most concerns were addressed. However, this reviewer still feels that the evidence linking PUB44-PBI and PBI1-WRKY45 in the chitin and PGN response is relatively weak. The authors explained that the antibodies against PUB44 and PBI1 did not work, which hampers their efforts in detecting the complex association in vivo. The authors have responded in the rebuttal letter that no paper showed the evidence that chitin responses occur in rice protoplasts. However, briefly checking research on the chitin perception system in rice, this reviewer has noticed reports that chitin could trigger the formation of OsLYP4/OsLYP6/OsCEBiP/OsCERK1 complexes and also the dissociation of OsCERK1 and OsRLCK176 in rice protoplast (Ao et al., 2014, The Plant Journal). Since PGN treatment works in rice protoplasts, the authors could also perform the complex formation assays using PGN. These experiments will explain and tighten up the relationship between OsPUB44, OsPBI1, and OsWRKY45 in chitin-triggered responses.

Answer: We thank for your suggestion. As pointed out by the reviewer, we have known that several papers used rice protoplasts-based transient expressing system for analyzing the protein-protein interaction. However, these papers did not show evidence that downstream immune responses occur in rice protoplast. Therefore, we developed the experimental system to monitor the interaction between PUB44 and PBI1 in rice protoplast during chitin and PGN responses. These experiments indicated that perception of PGN or chitin enhances the interaction between PUB44 and PBI1. We added these data in Fig. 1h,i. Since PBI1 degrades after chitin perception, we could not analyze the PAMP dependence of PBI1 – WRKY45 interaction.

(2) Figure 2F. The authors need to show the IP loading by α -Ubi western-blotting. Since they have the pbi1 mutant, they should include this as a negative control to confirm that the smear band is specific for ubiquitinated PBI1 in the Ubi assay.

Answer: According to the suggestion, we performed the co-immunoprecipitation assay with α -Ubiquitin using the *pbi1* mutant. The high-molecular weight bands of PBI1 was not observed in the *pbi1* mutant, indicating chitin-induced PBI1 ubiquitination. We added the data in Fig 2f.

(3) Title: delete "Precisely" and "master". This reviewer feels no evidence to support such claims from this work.

Answer: According to the suggestion, we deleted these words in the title.

Reviewer #3:

Having reviewed the resubmitted paper and the rebuttal I judge that the authors have responded to all of my comments satisfactorily.

Answer: We thank for the comments.

REVIEWERS' COMMENTS

Reviewer #1 (Remarks to the Author):

I appreciate the careful revision from the authors. The revision has addressed my comments and the manuscript is nicely presented. Congrats for the nice work in elucidating another branch of coordinated regulation by PBI1 and MAPKs on WRKYs.

Reviewer #2 (Remarks to the Author):

Authors have improved the writing and the connection between different sections has improved. The additional experiments have satisfactorily addressed my concerns.